# The role of somatosensory innervation of adipose tissues

Yu Wang[1,2], Verina H. Leung[1], Yunxiao Zhang[1,2], Victoria S. Nudell[1], Meaghan Loud[1,2], M. Rocio Servin-Vences[1,2], Dong Yang[1], Kristina Wang[1], Maria Dolores Moya-Garzon[3], Veronica L. Li[3], Jonathan Z. Long[3], Ardem Patapoutian[1,2 ✉] & Li Ye[1 ✉]

Adipose tissues communicate with the central nervous system to maintain whole-body energy homeostasis. The mainstream view is that circulating hormones secreted by the fat convey the metabolic state to the brain, which integrates peripheral information and regulates adipocyte function through noradrenergic sympathetic output[1]. Moreover, somatosensory neurons of the dorsal root ganglia innervate adipose tissue[2]. However, the lack of genetic tools to selectively target these neurons has limited understanding of their physiological importance. Here we developed viral, genetic and imaging strategies to manipulate sensory nerves in an organ-specific manner in mice. This enabled us to visualize the entire axonal projection of dorsal root ganglia from the soma to subcutaneous adipocytes, establishing the anatomical underpinnings of adipose sensory innervation. Functionally, selective sensory ablation in adipose tissue enhanced the lipogenic and thermogenetic transcriptional programs, resulting in an enlarged fat pad, enrichment of beige adipocytes and elevated body temperature under thermoneutral conditions. The sensory-ablation-induced phenotypes required intact sympathetic function. We postulate that beige-fat-innervating sensory neurons modulate adipocyte function by acting as a brake on the sympathetic system. These results reveal an important role of the innervation by dorsal root ganglia of adipose tissues, and could enable future studies to examine the role of sensory innervation of disparate interoceptive systems.

Mammalian adipose tissue is highly innervated. This innervation has been primarily studied for its efferent functions through tyrosine hydroxylase (TH)-expressing, noradrenaline-secreting sympathetic fibres[3,4]. These sympathetic fibres act on β-adrenergic receptors and have a well-recognized role in regulating thermogenesis and lipid metabolism in both brown and beige fat[5].

The afferent function of adipose innervation is much less understood. Adipose tissue is one of the few internal organs that receive little vagal sensory innervation[2]. By contrast, somatosensory fibres originating from the dorsal root ganglia (DRGs)—best known for skin and muscle sensation—have been reported to innervate adipose tissue in rats and hamsters[2,6], but the extent and importance of this innervation remain to be determined across species[7]. Although pioneering work using herpes viral tracing has elegantly mapped the central projection of fat afferent[8], the functional importance of this sensory innervation remains less understood because traditional chemical or surgical denervation of sensory fibres resulted in only minor phenotypes in the hamsters[2,9]. As we begin to appreciate the cellular and molecular heterogeneity of DRG neurons[10] and adipose tissue[5], the specificity of earlier classic denervation approaches needs to be re-evaluated. For example, whereas surgical denervation cannot discriminate between sensory and sympathetic fibres as they travel together in bundles, capsaicin denervation, which was thought to selectively ablate sensory fibres in fat, can target

non-neuronal transient receptor potential vanilloid (TRPV1)-expressing cells[11–13]. Moreover, capsaicin-mediated denervation is clearly biased towards the heat-sensitive and nociceptive TRPV1-expressing neurons (Aδ- and C-fibres)[14,15], potentially dampening the loss-of-function phenotypes[9]. By contrast, it is now known that a prominent non-peptidergic DRG population also expresses TH[16], calling into question the use of TH as a selective marker for sympathetic fibres in fat. These caveats motivated us to develop new imaging, molecular and circuit-specific tools to examine sensory innervation in adipose tissues.

## Direct visualization and characterization of the DRG projections to fat

In conventional tracing studies, viruses or dyes are injected into the fat to be retrogradely transported back to the DRG somas and assessed by histology. This indirect measurement is susceptible to varying tracer efficiency across hosts, potentially contributing to the discrepancy observed across animal species[6–8]. Ideally, direct visualization of the entire projection from DRG soma to the target organs, for example, through an axon-filling fluorophore, would provide the most reliable anatomical proof. However, the peripheral branch of the mouse DRG travels several centimetres before reaching its targets, making it impossible to visualize using conventional histology. We recently developed

[1]Department of Neuroscience, Dorris Neuroscience Center, Scripps Research, San Diego, CA, USA. [2]Howard Hughes Medical Institute, Chevy Chase, MD, USA. [3]Department of Pathology, Stanford School of Medicine, Sarafan ChEM-H, Stanford University, Stanford, CA, USA. ✉e-mail: ardem@scripps.edu; liye@scripps.edu

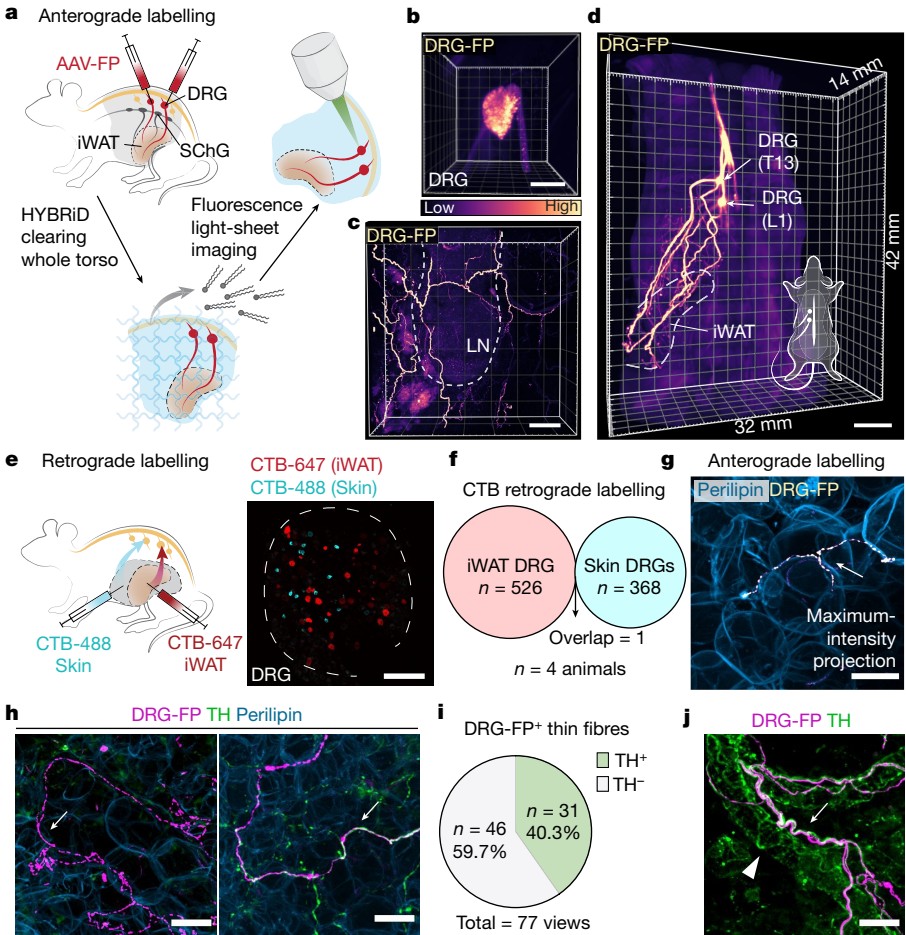

**Fig. 1 | Adipose tissues receive robust somatosensory innervations. a**, The workflow of mapping sensory innervation in adipose tissues. Tissues from mice with AAV expressing fluorescent protein (FP) injected in T13/L1 DRGs (vertebral level T13 and L1) were processed for en bloc HYBRiD clearing and fluorescence microscopy imaging. **b**–**d**, Representative 3D image volume of AAV-labelled DRGs (T13) by light-sheet imaging (**b**), DRG fibres in the iWAT by confocal imaging (showing lymph node (LN) as a landmark) (**c**) and DRG axonal projections in an adult mouse torso (**d**). The colour gradient suggests relative intensity. LN, lymph node. **e**, Schematic of dual-colour CTB labelling from iWAT and flank skin (left) and representative whole-mount image of DRGs (right,

T13). **f**, Quantification of CTB-positive cell numbers from labelling in the iWAT and skin. $n = 4$ mice. **g**, Representative image of virally labelled DRG fibre in close apposition to an adipocyte. **h**, Representative image of TH⁻ and TH⁺ parenchymal DRG innervation. **i**, Quantification of the percentage of TH⁺ DRG fibres (77 views in 13 images from 3 biological samples). **j**, Representative image of virally labelled DRG fibre travelling along vasculature. The white arrows mark DRG fibres, and the white triangle marks TH-stained sympathetic fibres. Scale bars, 200 μm (**b** and **e**), 500 μm (**c**), 5 mm (**d**) and 30 μm in (**g**, **h** and **j**).

HYBRiD, specifically designed for en bloc fluorescence visualization of large tissues[17], paving the way to directly characterize the intact sensory innervation from DRGs to fat.

Another barrier to selectively labelling sensory innervation in the fat is that common pan-DRG Cre transgenic mice (such as Pirt-Cre, Scn10a-Cre, Advillin-CreERT2[18]) also target sympathetic neurons (Extended Data Fig. 1a). We therefore adopted an intraganglionic DRG surgery in mice to directly inject a recombinant adeno-associated virus (AAV) expressing fluorescent protein into individual DRGs without transducing sympathetic ganglia (Fig. 1a and Extended Data Fig. 1b). For the rest of the study, we focused on the inguinal white adipose tissue (iWAT), a beige fat pad with well-established roles in mouse physiology[19]. We injected a fluorescent-protein-expressing AAV into the thoracolumbar DRGs (vertebral level T13 and L1 (T13/L1)[6]), to target all projecting fibres from these two ganglia. After en bloc HYBRiD clearing and light-sheet imaging of the whole torso (Fig. 1a), the entire projection from DRG soma to iWAT travelling across 1.2 cm can be resolved (Fig. 1b–d and Supplementary Video 1), unequivocally demonstrating that thoracolumbar DRGs directly innervate iWAT (Extended Data Fig. 1c).

Furthermore, retrograde cholera toxin subunit B (CTB) labelling was also used to confirm the imaging findings. As expected[6,8,20,21], iWAT receives sensory innervation and sympathetic innervation mainly from T11–L3 DRGs and paravertebral sympathetic chain ganglia (SChGs), respectively; but not from nodose ganglion (vagal nerve) or collateral sympathetic ganglia (Extended Data Fig. 2a,b). Fat-projecting DRGs consist of multiple neuron types with an enrichment of peptidergic and myelinated fibres (Extended Data Fig. 2h,i). Given the proximity between iWAT and skin (the primary DRG target), we examined whether fat innervation is part of the cutaneous sensory circuit. Dual-colour CTB labelling from iWAT and neighbouring flank skin showed that these two organs are innervated by two non-overlapping DRG populations (Fig. 1e,f). Visceral fat (epididimal white adipose tissue (eWAT)) also receives sensory innervation at comparable vertebral levels but from a separate population (Extended Data Fig. 2d,e). Together, these 3D imaging and retrograde labelling data demonstrate that distinct neurons in the thoracolumbar DRGs robustly project to adipose tissues.

The volumetric images enabled us to examine the morphology and topology of somatosensory terminals in adipose tissue in a high level of detail (Fig. 1g–j). The sensory fibres in iWAT can be divided into at

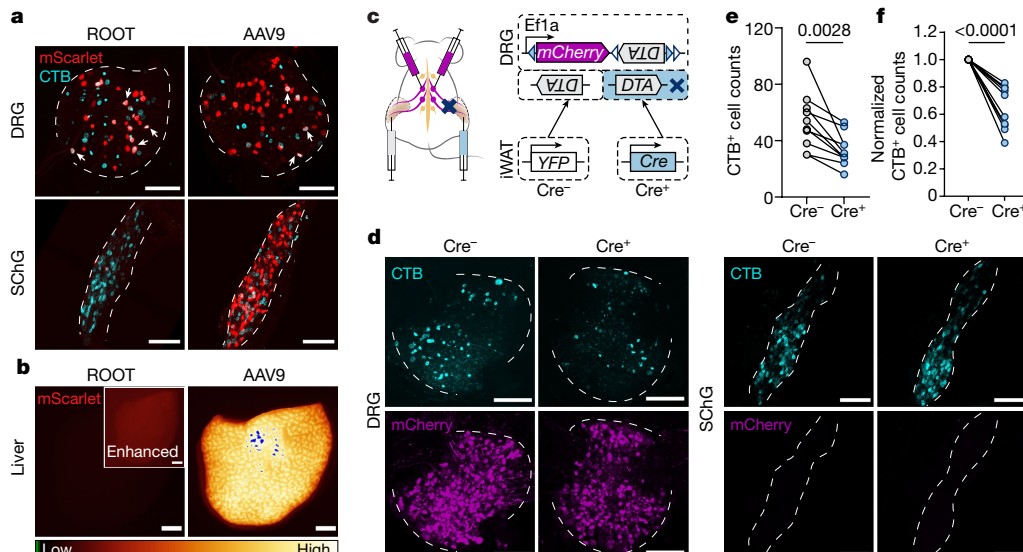

**Fig. 2 | Combinatorial strategy for specific iWAT-DRG manipulation.**
**a**,**b**, Comparison of ROOT and AAV9 for retrograde labelling in the iWAT.
**a**, Representative whole-mount images of DRGs (T13) and SChGs (T12) labelled by ROOT-mScarlet or AAV9-mScarlet and a sequential CTB-647 injection from iWAT. Some of the AAV and CTB double-positive cells are highlighted by white arrows. CTB labels 26.42 ± 6.49% of ROOT-labelled neurons, and 9.95 ± 0.92% of AAV9-labelled neurons in T13/L1 DRGs. Data are mean ± s.e.m. $n$ = 4 mice per group. **b**, Representative images of livers from animals with ROOT-mScarlet or AAV9-mScarlet injected into the iWAT. **c**–**f**, Combinatorial viral strategy for Cre-dependent ablation of iWAT DRGs. **c**, Schematic of the combinatorial viral

strategy for unilateral sensory ablation. Each mouse has AAV-mCherry-flex-DTA injected into the bilateral T13/L1 DRGs, and ROOT-YFP or ROOT-Cre injected into the iWAT unilaterally. Three weeks after surgery, CTB-647 was injected into the iWAT bilaterally. **d**, Representative whole-mount images of contralateral (Cre⁻) and ipsilateral (Cre⁺) DRGs (T13) and SChGs (T12).
**e**,**f**, Quantification of CTB⁺ cell numbers (**e**) and normalized cell numbers (**f**) in T13 and L1 DRGs labelled from iWAT. $n$ = 10 mice. Statistical analysis was performed using two-tailed paired $t$-tests. $P$ values are shown at the top. For **a**,**b** and **d**, scale bars, 200 μm.

least two main types: (1) larger bundles travelling along the vasculature (Fig. 1j and Extended Data Fig. 3e) and (2) parenchymal innervation, in which the sensory nerve terminates in close apposition to adipocytes (Fig. 1g,h and Extended Data Fig. 3a,d). TH has long been regarded as a sympathetic marker and a surrogate for adipose sympathetic innervation[4,7]. For the larger bundles, sensory fibres travel together with TH⁺ sympathetic fibres along the vasculature but rarely wrap the vessel as the latter typically do[4] (Fig. 1j and Extended Data Fig. 3e). Notably, in the parenchymal portion, nearly 40% of thin sensory terminals close to adipocytes are immunopositive for TH (Fig. 1h,i and Extended Data Fig. 3d), challenging the traditional view that TH is an exclusive sympathetic marker in fat and, importantly, suggesting that earlier studies based on TH might be confounded, at least partially, by the sensory innervation. Thus, with these findings in mind, it is imperative to establish the specific functions of sensory innervation in adipose tissue.

## Selectively targeting adipose sensory innervation

Sympathetic output in adipose tissues works through β-adrenergic receptors, enabling the use of catecholaminergic neurotoxins (such as 6-hydroxydopamine (6-OHDA)) and adrenergic ligands to specifically manipulate their activities. By contrast, sensory fibres are more diverse and respond to different sensory modalities and therefore cannot be manipulated based on a single signalling pathway. Thus, a neuron-specific, projection-defined genetic approach is necessary to study sensory innervation in adipose tissue.

A combination of axonal target injection of retrograde Cre and somatic expression of a Cre-dependent payload has been widely used to manipulate projection-specific circuits in the brain. However, legacy peripheral viral tracers such as pseudorabies virus and herpes simplex virus are highly toxic, restricting their use beyond acute anatomical mapping. In search for newer and safer viral vectors suitable for long-term functional manipulations, we found that AAV9 exhibited a

high retrograde potential from iWAT to DRGs (Extended Data Fig. 4a). We adopted a published viral engineering pipeline[22] to generate randomized mutants of AAV9 (Extended Data Fig. 4b,c). Although our initial intent was to improve retrograde efficiency from fat to DRGs, the evolved new retrograde vector optimized for organ tracing (or ROOT) is more desirable mainly for its significantly reduced off-target expression such as in SChGs, contralateral DRGs and the liver (Fig. 2a,b and Extended Data Fig. 4d–g).

ROOT provides an opportunity to specifically ablate the sensory innervation in fat—we injected Cre-dependent diphtheria toxin subunit A (DTA) construct (mCherry-flex-DTA) into the T13/L1 DRGs bilaterally while injecting Cre- or YFP-expressing ROOT unilaterally in the iWATs (Fig. 2c). On the basis of CTB quantification, we noticed a decrease of approximately 40% in fat-projecting neurons in Cre⁺ ipsilateral DRGs compared with the control side (Fig. 2d–f), comparable to the efficiency achieved by previous viral DTA ablations[23]. No difference was observed in the SChGs (Fig. 2d and Extended Data Fig. 5a). Importantly, the structure and function of sensory terminals in the flank skin remained intact (Extended Data Fig. 5b–e). Together, our projection-defined, partial sensory ablation in the iWAT provides a specific loss-of-function model to study the adipose-innervating DRGs.

## Fat gene program changes after sensory ablation

We next investigated how the loss of sensory innervation affects the molecular programs of the iWAT. We compared the sensory-ablated iWAT (Cre⁺) with the non-ablated contralateral fat pad (YFP⁺) within the same animal. RNA-sequencing (RNA-seq) analysis was performed in the fat pads 3–4 weeks after viral injection (Fig. 3a–d). Unbiased Gene Ontology analysis showed that both fatty acid and lipid metabolism and cold-induced thermogenesis pathways (Fig. 3d) were enriched by sensory ablation. At the individual-gene level, well-established thermogenic brown/beige cell markers were significantly upregulated

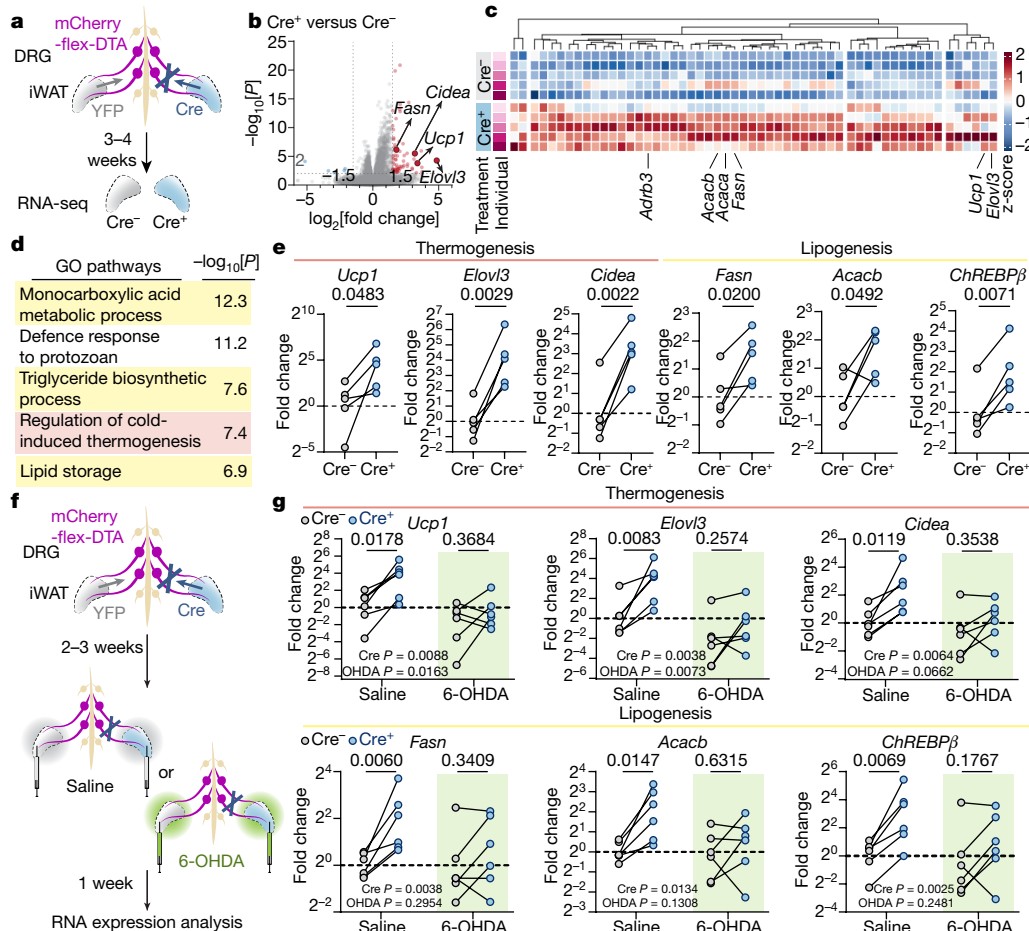

**Fig. 3 | Ablation of iWAT DRGs upregulates thermogenic and lipogenic transcriptional programs. a–e**, Transcriptional profiling of the iWAT after Cre-dependent sensory ablation. **a**, Schematic of transcriptional profiling. The iWAT from mice with Cre-dependent unilateral sensory ablation was processed for RNA-seq analysis. **b**, RNA-seq results. Statistical analysis was performed using Wald tests. **c**, Heat map of upregulated genes identified by RNA-seq analysis. **d**, Gene Ontology (GO) enrichment analysis of upregulated genes. **e**, qPCR with reverse transcription (RT–qPCR) analysis of thermogenic and lipogenic genes in the iWAT after unilateral sensory ablation. $n = 5$ mice.

Statistical analysis was performed using two-tailed paired $t$-tests. **f**,**g**, Transcriptional analysis of iWAT with sensory ablation and sympathetic ablation. **f**, Schematic of sensory and sympathetic double ablation. Mice with Cre-dependent unilateral sensory ablation were subjected to bilateral sympathetic 6-OHDA denervation. **g**, RT–qPCR analysis of thermogenic and lipogenic genes in the iWAT with or without sympathetic denervation. $n = 6$ mice per group. Statistical analysis was performed using two-way analysis of variance with Sidak's multiple-comparisons test.

in the ablated side, including *Ucp1* (10.4-fold, $P = 1.6 \times 10^{-4}$), *Elovl3* (28.8-fold, $P = 5.2 \times 10^{-5}$) and *Cidea* (9.1-fold, $P = 3.3 \times 10^{-6}$). Markers for de novo lipogenesis (DNL)[24], such as *Fasn* (3.5-fold, $P = 7.1 \times 10^{-7}$) and *Acacb* (2.6-fold, $P = 1.1 \times 10^{-4}$), also showed elevated expression (Fig. 3b,c). The concurrent activation of opposing lipid oxidation and lipogenesis pathways is a unique but well-documented mechanism in the adipose tissue to ensure fuel availability for heat production under cold or β-adrenergic stimulation[25,26]. The mRNA expression of these genes, together with that of the gene encoding ChREBPβ (*Mlxipl*; called '*ChREBPβ*' here), a transcriptional master regulator of DNL[27,28], was confirmed by quantitative PCR (qPCR), showing increases in the ablated iWAT (Fig. 3e) but not in non-targeted eWAT or interscapular brown adipose tissues (iBAT) (Extended Data Fig. 6a,b).

As thermogenic and lipogenesis programs are both downstream of sympathetic signalling[24,26,29], we next tested whether the sensory-elicited gene expression changes are dependent on intact sympathetic innervation. We bilaterally injected 6-OHDA—a catecholaminergic toxin for selective sympathetic denervation (Extended Data Fig. 6c,d)—into the iWAT of mice that had previously undergone unilateral sensory ablation (Fig. 3f). This double-ablation experiment showed that the induction of thermogenic and lipogenic genes by

sensory ablation was blunted in 6-OHDA-treated animals (*Ucp1*: 9.4-fold (saline) to 2.6-fold (6-OHDA); *Elovl3*: 12.0-fold to 2.9-fold; *Cidea*: 4.7-fold to 1.8-fold; *Fasn*: 3.5-fold to 1.6-fold; *Acacb*: 3.6-fold to 1.4-fold; *ChREBPβ*: 6.7-fold to 2.5-fold) (Fig. 3g and Extended Data Fig. 6e,f), indicating that sensory-regulated gene expression changes are at least partially dependent on the intact sympathetic function.

## Sensory regulation of adipose physiology

We next examined how the sensory-ablation-induced gene changes affect adipose functions (Fig. 4a–c). We observed higher phosphorylation of hormone-sensitive lipase (HSL) and enrichment of multilocular beige adipocytes in the unilateral sensory-ablated iWAT (Fig. 4c and Extended Data Fig. 7d–f), consistent with upregulation of the thermogenic program (Fig. 3). This resembles the beiging of iWAT after cold exposure or β-adrenergic agonism. However, under these conditions, the wild-type fat pads normally shrink in size, presumably due to stronger lipid utilization than DNL. By contrast, we observed an increase in fat mass of the sensory-ablated iWAT compared with the contralateral controls (Fig. 4a,b and Extended Data Fig. 7a–c), suggesting that the sensory innervation not only counteracts sympathetic

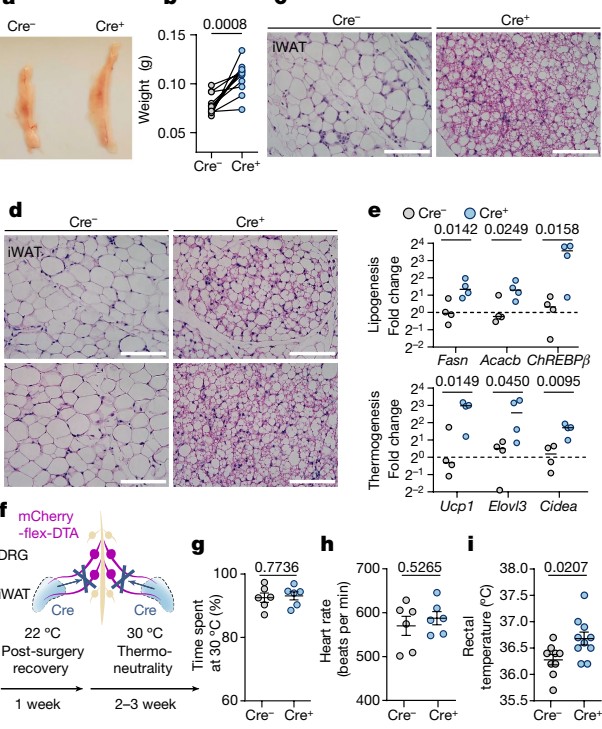

**Fig. 4 | Ablation of iWAT DRGs changes the morphological and physiological properties of the iWAT. a,b**, Representative images (**a**) and quantification of fat mass (**b**) of iWAT with Cre-dependent unilateral sensory ablation. $n = 11$ mice. Statistical analysis was performed using two-tailed paired $t$-tests. **c**, Histology of iWAT with Cre-dependent unilateral sensory ablation. **d**, Histology of iWAT with Cre-dependent bilateral sensory ablation. Each panel is from a different mouse. **e**, RT–qPCR analysis of iWAT with Cre-dependent bilateral sensory ablation. $n = 4$ mice per group. Statistical analysis was performed using two-tailed unpaired $t$-tests. **f–i**, Physiological measurements after Cre-dependent bilateral sensory ablation. **f**, Schematic of bilateral sensory ablation, and the timeline of physiological measurement. **g**, Two-temperature choice assay (30 °C versus 18 °C) of Cre⁻ ($n = 6$) and Cre⁺ ($n = 6$) mice. **h**, Heart rate of Cre⁻ ($n = 6$) and Cre⁺ ($n = 6$) mice. **i**, Rectal temperature at thermoneutrality of Cre⁻ ($n = 9$) and Cre⁺ ($n = 10$) mice. For **g–i**, data are mean ± s.e.m. Statistical analysis was performed using two-tailed unpaired $t$-tests with Welch's correction (**g–i**). For **c** and **d**, scale bars, 100 µm.

activity, but may also have a role in coordinating these two opposing downstream pathways.

Unilateral ablation enabled accurate assessment of adipose phenotypes within the same animal. To determine how adipose sensory innervation affects whole-body physiology, we further ablated sensory innervation in the iWAT bilaterally. Consistent with the unilateral ablation, we observed an enrichment of beige adipocytes (Fig. 4d) and upregulated lipogenic and thermogenic genes in iWAT in the ablated animals (Fig. 4e and Extended Data Fig. 8a,b). These animals were housed at murine thermoneutrality (30 °C)[30] (Fig. 4f) to eliminate confounding effects from other thermoregulatory mechanisms (such as background brown fat activity). Fat sensory ablation did not lead to significant changes in body weight, food intake, temperature sensitivity or systemic sympathetic tone (Fig. 4g,h and Extended Data Fig. 8c–i), but exhibited elevated core body temperature compared with the controls (Fig. 4i), consistent with an enhanced thermogenesis program in the iWAT. Interestingly, the difference in body temperature was normalized at 22 °C (Extended Data Fig. 8c), suggesting that the elevated body temperature at thermoneutrality was not due to a deficit in central thermoregulatory mechanisms (pyrexia)[31,32]. After challenge

with a high-fat diet, despite showing a small difference in body weight, the mice with adipose sensory ablation exhibited markedly improved glucose tolerance compared with the control mice (Extended Data Fig. 8j–n). Such disproportional changes in glucose tolerance and body weight resemble the phenotype reported in PRDM16-induced beiging in transgenic models[33,34], suggesting that adipose sensory ablation could protect mice from diet-induced glucose intolerance through beige-fat-related mechanisms.

## Discussion

The conventional view posits that adipose signals transmit to the brain through slow, diffusive circulating hormones. Here the unequivocal anatomical proof of sensory innervation in fat provides a circuitry basis for a potential fast, spatially encoded neural transmission from the peripheral organs to the brain. Indeed, we go on to show evidence that DRG sensory neurons act as an inhibitory break on the local sympathetic function, reminiscent of the role of vagal baroreceptors in modulating blood pressure[35,36]. These findings fill an important gap of how the central nervous system monitors and orchestrates adipose functions, highlighting the importance of an underappreciated branch of brain–body communication.

It remains to be determined how sensory pathways mechanistically interact with sympathetic signalling. It has been suggested that capsaicin denervation of iWAT altered sympathetic output in the distal iBAT through central circuits[8,37]; however, this observation might be confounded by the likely increase in sympathetic tone[38] due to peripheral capsaicin administration. By contrast, with projection-specific ablation, our adipose phenotypes were strictly limited in the ipsilateral iWAT but not in the distal iBAT or eWAT (Figs. 3 and 4 and Extended Data Figs. 5 and 6), suggesting a local specificity of the sensory–sympathetic interaction (Fig. 3f,g). Sensory modulation of sympathetic activity has been documented in other organs[39,40] at the spinal or supraspinal levels[41–43]. We did not observe a significant change in total noradrenaline content in sensory-ablated fat (Extended Data Fig. 8i), suggesting that sensory activity may act on sympathetic signalling downstream of the adrenergic receptors, although we cannot rule out potential temporospatial noradrenaline changes, which cannot be resolved by a single, bulk noradrenaline measurement. Moreover, remodelling of sympathetic fibres is a hallmark of adipose adaptation to chronic metabolic challenges, and it remains to be tested whether sensory innervation undergoes a similar process, especially in different strains of mice as well as across species.

Our study raises many questions. The somatosensory nervous system, which is characterized by clusters of first-order neurons that mainly reside in the DRG, has great molecular heterogeneity as profiled by recent single-cell transcriptomes[10,44]. Which DRG subtypes innervate fat and whether distinct subtypes have different functions (that is, regulating thermogenesis versus lipogenesis) is currently unclear. This could potentially be determined in the future by coupling ROOT-based retrograde targeting with single-cell RNA-seq to establish the identities the fat-innervating neurons. This information could also help to identify the interoceptive signal that fat-innervating neurons sense. Previous research showed that infusion of exogenous leptin and free fatty acids could activate DRGs by FOS staining or ex vivo recording[45,46]; however, the identification of endogenous signals (chemical or physical) will probably require a full understanding of the putative receptor expression using the organ-targeted single-cell RNA-seq approach suggested above. Moreover, the nature of neural transmission also suggests that these endogenous activities probably occur at second or millisecond scales. Thus, a matching ability to readout projection-specific DRG activities in vivo would be required to identify the triggering signals, for example, using emerging long-term DRG calcium imaging techniques[47,48].

Our research also has implications beyond the sensory innervation of fat. Interoception has been mostly studied from the perspective of cranial ganglia of vagal origin[49]. Increasing evidence indicates the DRG innervation of internal organs also has a crucial role in interoception[50]. The discoveries made here could serve as a proof-of-principle for investigating the role of DRG neurons in a variety of other internal organs.

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

# Methods

## Animals

Mice were group housed in standard housing under a 12–12 h light–dark cycle with ad libitum access to chow diet and water, with the room temperature kept around 22 °C and humidity kept between 30–80% (not controlled). Mice were single housed for food intake measurement and held at 30 °C for thermoneutral exposure experiments. Mice (aged at least 6 weeks) from the following strains were used for this study: wild-type (WT) C57BL/6J (Jackson stock, 000664), B6.Cg-Gt(ROSA)2 6Sor^tm9(CAG-tdTomato)Hze/J (Jackson, 007909, Ai9), Pirt-Cre[51], Scn10a-Cre[52]. Both genders were used for anatomical mapping studies, and male mice were used for *in vivo* functional experiments. All of the experimental protocols were approved by The Scripps Research Institute Institutional Animal Care and Use Committee and were in accordance with the guidelines from the NIH.

## Viruses

CAV2 was obtained from CNRS Vector Core. rAAVretro-Cre was obtained from Boston Children's Hospital and Janelia. PHP.S-Cre, PHP.S-DIO-sfGFP (gift from D. Gibbs) was obtained from Janelia. PHP.S-TdTomato (Addgene, 59462), PHP.S-mScarlet (gift from the Deisseroth laboratory and cloned in-house), ROOT-Cre (Addgene, 51904), ROOT-EYFP (cloned in-house), PHP.S-mCherry-flex-DTA (Addgene, 58536), ROOT-mScarlet, AAV9-mScarlet and PHP.S-mScarlet were packaged in-house using published protocol[53]. AAVs produced in-house were titrated by qPCR before aliquoting into 6–10 μl and flash-frozen for long-term storage.

## In vivo selection of ROOT

**Plasmids.** The plasmids used for in vivo selection were adapted from the previous publication[54]. rAAV-pUBC-sfGFP-Cap and AAV2/9-REP-AAP was generated from pUBC-mCherry-rAB (Addgene, 115239), pUCmini-iCAP-PHP.S (Addgene, 103006), pAAV2/8 (Addgene, 112864). No in-cis-Lox module or transgenic Cre lines were used given the anatomical separation of DRGs and iWAT. The ROOT capsid library was generated by inserting random heptamers using NNK degenerate primers (Integrated DNA technologies) between the 588 and 589 sites of AAV9 by Gibson assembly as previously described[54].

**AAV capsid library production.** The viral libraries were produced as previously described[53,54]. In brief, only 10 ng of rAAV-pUBC-sfGFP-Cap library DNA was transfected in HEK293FT (Invitrogen R70007) cells per 150 mm plate, and the virus was collected after 60 h for purification.

**DNA recovery and sequencing.** The resulting AAV capsid library was injected bilaterally into the iWAT of C57Bl/6J male mice at $10^9$ viral genomes (VGs) per fat pad. rAAV genomes were recovered from T12–L3 DRGs from injected mice two weeks after injection using the DNeasy Blood and Tissue kit (Qiagen). The recovered viral DNA was amplified for two rounds against a fragment containing the heptamer insertion with Q5 high-fidelity polymerase (New England Biolabs). The amplified products were cleaned up and processed for complete amplicon sequencing at Massachusetts General Hospital.

**NGS data alignment and processing.** Raw FASTQ files from NGS runs were aligned to an AAV9-template DNA fragment containing the 21 bp diversified region between amino acids 588 and 589 using SAMtools (v.1.10). The abundance of each 21 bp sequence in all of the recovered sequences with the heptamer insertion was quantified.

## Surgery

Mice were anaesthetized with isoflurane (4% for induction and 1.5–2% for maintenance), with their skin at the surgical area shaved and hair removed and sterilized using ethanol and iodine. After surgery, the mice were given a subcutaneous injection of flunixin and topical antibiotic ointment for post-operative care.

**Retrograde tracer labelling of sensory neurons.** For injection into the iWAT, a lateral incision was made on the flank skin of each side. For injection into the eWAT, a lateral incision was made on the lower abdominal wall. For injection into the iBAT, a midline incision was made in the interscapular region. For injection into the skin, an intradermal injection was performed. A Hamilton syringe with a 31G (point type 2) needle was used for all retrograde tracing. A total of 4–5 μl 0.1% CTB-488 or CTB-647 (Invitrogen) in PBS was injected per fat pad (2 μl for iBAT) or the flank skin with 8–15 injections to spread out the tracer. The abdominal wall (for eWAT injection) and skin (for iWAT, eWAT or iBAT injection) were sutured separately. Tissues were taken 3–5 days after injection to allow the dye to reach the DRG soma.

**Retrograde viral labelling of sensory neurons.** Injections were performed as described above. For the viral comparison study (Extended Data Fig. 3a), CAV2-Cre ($4.2 \times 10^{12}$ physical particles per ml, 5 μl), PHP.S-Cre ($1.6 \times 10^{13}$ VGs per ml, 3 μl), rAAVretro-Cre ($9 \times 10^{12}$ VGs per ml, 5 μl) was mixed with 0.01% FastGreen (Sigma-Aldrich) and unilaterally injected into iWAT of Ai9 mice. AAV9-TdTomato ($2 \times 10^{13}$ VGs per ml, 2 μl) was mixed with 0.01% FastGreen (Sigma-Aldrich) and unilaterally injected into iWAT of WT mice.

**Characterization of ROOT.** Injections were performed as described above. For ROOT characterization, AAV9 or ROOT-mScarlet ($4 \times 10^{13}$ VGs per ml, 2 μl) in PBS with 0.001% F-68 and 0.01% FastGreen was administered unilaterally into iWAT of WT mice. Two weeks after the first surgery, 4 μl of 0.1% CTB-647 was injected into the same iWAT fat pad, and tissues were taken 3–5 days after the second injection for quantification.

**6-OHDA treatment.** 6-OHDA has been used previously for selective sympathetic denervation[55,56]. 6-OHDA (Tocris) (12 mg ml$^{-1}$ in saline with 0.02% ascorbic acid (Sigma-Aldrich)) was prepared fresh before use and kept on ice and in the absence of light. 6-OHDA (8 μl) was injected into each iWAT fat pad. Saline with 0.02% ascorbic acid was used as a control. Tissues were taken 7–9 days after injection.

**Intraganglionic DRG injection.** Intraganglionic DRG injection was performed according to a previous report[57]. A midline incision was made on the dorsal skin to expose the dorsal muscles. The muscles along the vertebra were carefully separated to expose DRGs. DRGs at the T13 and L1 vertebral level were exposed, and AAV (~1 × 10$^{13}$ VGs per ml in PBS with 0.001% F-68 and 0.01% FastGreen) was injected in the ganglion (~200 nl per ganglion) with a pulled glass pipette using the Nanoliter 2020 Injector (World Precision Instrument). Care was taken to avoid damaging the surrounding vasculatures. Dorsal muscle and skin were sutured separately. PHP.S-TdTomato or PHP.S-mScarlet was used for anterograde labelling.

**Cre-dependent ablation of iWAT-innervating DRGs.** To selectively ablate iWAT-innervating DRGs, ROOT-Cre or ROOT-YFP ($4 \times 10^{13}$ VGs per ml, 2 μl) in PBS with 0.001% F-68 and 0.01% FastGreen was injected into iWAT unilaterally or bilaterally, and PHP.S-mCherry-flex-DTA ($1 \times 10^{13}$ VGs per ml, 200 nl per ganglion) was injected into T13 and L1 DRGs bilaterally as described above. Tissues were extracted, or physiological measurements were performed 3–4 weeks after surgery.

**6-OHDA treatment and Cre-dependent sensory ablation.** AAVs were injected into iWAT and DRGs to achieve Cre-dependent sensory ablation as described above. Two to three weeks after the first injection, 6-OHDA or saline was injected into the iWAT again. Tissues were extracted 7–9 days after the second injection.

**En bloc HYBRiD tissue clearing and immunolabelling of iWAT**
To visualize sensory nerves in mouse torso and iWAT samples, tissue samples were cleared using the HYBRiD method as described previously[17].

**Sample collection and pretreatment.** Mice were terminally anaesthetized with isoflurane and intracardially perfused with ice-cold PBS and ice-cold 4% PFA in PBS with 4% sucrose (Electron Microscopy Perfusion Fixative, 1224SK). For torso samples, the skin was carefully removed leaving the iWAT attached to the muscle, the spinal cord was cut from the midline to facilitate clearing and imaging. All of the collected samples were post-fixed in 4% PFA at 4 °C for 1–2 days before being washed in PBS. The torso samples were decalcified in 10% EDTA/15% imidazole (at 4 °C for 7 days), then decolourized in 25% N,N,N′,N′-tetrakis(2-hydroxypropyl)ethylenediamine (Quadrol) (in 1× PBS) (at 37 °C for 4 days).

**Delipidation and hydrogel embedding.** The torso and iWAT samples were sequentially washed in 50%, 70%, 80%, 95% and 95% tetrahydrofuran in 25% Quadrol (in 1× PBS), 100% dichloromethane (DCM), 100% DCM, 100% DCM, 95%, 95%, 80%, 70% and 50% tetrahydrofuran in 25% Quadrol followed by 1× PBS to wash out any remaining organic solvent. The samples were then incubated in A1P4 hydrogel (1% acrylamide, 0.125% Bis, 4% PFA, 0.025% VA-044 initiator (w/v), in 1× PBS) at 4 °C before being degassed with nitrogen and polymerized (4 h at 37 °C). The samples were then removed from the hydrogel and passively cleared with 20 mM LiOH·Boric buffer pH 8.0 containing 6% SDS at 37 °C until the samples appeared translucent. After clearing, the samples were washed in PBST (0.2% Triton X-100) thoroughly before proceeding to refractive-index matching or immunolabelling.

**Immunolabelling.** iWAT samples were incubated with primary antibodies diluted in PBST at room temperature. After incubation with primary antibodies, the samples were washed in PBST and then incubated in secondary antibodies diluted in PBST at room temperature. The samples were washed extensively in PBST before refractive-index matching and mounting. The following antibodies were used: anti-perilipin-1 (Cell Signaling, 9349, 1:400), anti-TH-647 (BioLegend, 818008, 1:300); anti-Ucp1 (Abcam, ab10983, 1:200); anti-rabbit-488 (Jackson Immuno Research 711-546-152, 1;400); anti-rabbit-647 (Jackson Immuno Research 711-606-152, 1:400).

**Refractive-index matching and mounting.** Cleared or immunolabelled samples were refractive-index matched in EasyIndex (RI = 1.52, Life Canvas) and mounted in spacers (Sunjin Lab) for confocal microscopy imaging or mounted in agarose for light-sheet microscopy imaging.

**Histological analysis of whole-mount samples and cryosections**
Mice were terminally anaesthetized with isoflurane and intracardially perfused with PBS and 4% PFA. For flank skin samples, skin was shaved, and hair was removed using Nair.

**Whole-mount imaging of ganglia and flank skin.** Ganglia of interest (DRGs, SChGs and celiac/mesenteric complex) were dissected and mounted in RapiClear (Sunjin Lab) with silicone spacer (Electron Microscopy) for confocal imaging. Flank skin was dissected, post-fixed in PFA and washed three times in PBS before mounting in Fluoromount-G (Invitrogen 00-4958-02) using 0.25 mm iSpacers (Sunjin Lab) for confocal imaging.

**Immunolabelling analysis of skin sections.** Flank skin was dissected, post-fixed in PFA, dehydrated in 30% sucrose before being embedded in OCT, then sectioned at 25 µm and mounted on gelatin-coated slides. For immunofluorescence analysis, skin tissue sections were blocked with 5% normal donkey serum in PBS with 0.3% Triton X-100. Primary antibodies were prepared in the same blocking solution and incubated overnight (anti-βIII-tubulin (Abcam, ab18207, 1:1,000). The next day, the sections were washed in PBS and then incubated for 2 h at room temperature with secondary antibodies (anti-rabbit-647, Jackson Immuno Research, 711-606-152, 1:500), and stained with DAPI before mounted with ProLong Gold antifade mountant (Invitrogen) for confocal microscopy imaging.

**Immunolabelling of DRG sections.** T12 to L2 DRGs from mice with CTB-647 injected in iWAT were dissected, post-fixed in PFA, dehydrated in 30% sucrose before being embedded in OCT, then sectioned at 20 µm. DRG sections were stained according to the procedure described above using the following primary antibodies: anti-CGRP (Immunostar, 24112, 1:1,000), anti-neurofilament heavy polypeptide (Abcam, ab4680, 1:1,000). Secondary antibodies and dyes included anti-rabbit-594 (Jackson Immuno Research, 711-586-152, 1:500), anti-chicken-488 (Jackson Immuno Research, 703-546-155, 1:500) and isolectin B4 AlexaFluor 488 (Life Technologies, I21411, 25 µg ml⁻¹).

**Haematoxylin and eosin staining of adipose tissues.** Mice were terminally anaesthetized with isoflurane. iWAT, eWAT and iBAT were extracted and post-fixed in 4% PFA overnight at 4 °C. The tissues were paraffin-embedded, sectioned at 5 µm and mounted onto glass slides. The sections were then stained with haematoxylin and eosin at Sanford Burnham Prebys histology core and imaged using a bright-field microscope.

## Imaging

**Confocal microscopy.** Mounted iWAT samples, whole-mount ganglia, stained skin sections were imaged with Olympus FV3000 confocal microscope using one of the following objectives: ×4/0.28 NA, air (XLFluor, Olympus); ×10/0.6 NA, water immersion (XLUMPlanFI, Olympus). Images were acquired using Fluoview (v.2.4.1.198).

**Light-sheet microscopy.** Torso or iWAT samples were mounted using 1% agarose/EasyIndex. Mounted samples were imaged inside the SmartSPIM chamber filled with EasyIndex and sealed with mineral oil on the top. Images were acquired using a ×3.6/0.2 NA objective (LifeCanvas), with a 1.79 µm, 1.79 µm, 4 µm $xyz$ voxel size. Image acquisition was completed with bilateral illumination along the central plane of symmetry within the sample.

**Fluorescence stereomicroscopy.** Freshly fixed liver samples were imaged using the Leica M165 FC stereomicroscope and FLIR BFS-U3-51S51M-C camera. Images were acquired using SpinView (v.2.5.0.80).

**Bright-field microscopy.** Slides stained with haematoxylin and eosin were imaged using the Keyence BZ-X710 microscope with a ×40/0.6 NA objective (CFI S Plan Fluor ELWD ADM, Nikon).

## Transcriptional analysis

**RNA preparation and RT–qPCR analysis.** Adipose tissues were dissected between 12:00 and 14:00 and flash-frozen in liquid nitrogen. Total RNA was extracted from frozen tissue using TRIzol (Invitrogen) and RNeasy Mini kits (Qiagen). For RT–qPCR analysis, total RNA was reverse-transcribed using the Maxima H Minus First Strand cDNA Synthesis Kit (Thermo Fisher Scientific). The resultant cDNA was mixed with primers (Integrated DNA Technology) and SyGreen Blue Mix (Genesee Scientific, 17-507) for RT–qPCR using the CFX384 real-time PCR system (BioRad). Normalized mRNA expression was calculated using $\Delta\Delta C_t$ method, using *Tbp* (encoding TATA-box-binding protein) mRNA as the reference gene. Statistical analysis was performed on $\Delta\Delta C_t$. The primer sequences (forward and reverse sequence, 5′ to 3′, respectively) were as follows: *Tbp* (CCTTGTACCCTTCACCAATGAC and

ACAGCCAAGATTCACGGTAGA); *Ucp1* (AGGCTTCCAGTACCATTAGGT and CTGAGTGAGGCAAAGCTGATTT); *Elovl3* (TTCTCACGCGGGT-TAAAAATGG and GAGCAACAGATAGACGACCAC); *Cidea* (ATCACAACTG-GCCTGGTTACG and TACTACCCGGTGTCCATTTCT); *Fasn* (GGAGGTG-GTGATAGCCGGTAT and TGGGTAATCCATAGAGCCCAG); *Acacb* (CGCT-CACCAACAGTAAGGTGG and GCTTGGCAGGGAGTTCCTC); *ChREBPβ* (TCTGCAGATCGCGTGGAG and CTTGTCCCGGCATAGCAAC).

**RNA library preparation and sequencing.** RNA library preparation and sequencing was performed at Scripps Genomics core. Total RNA samples were prepared into RNA-seq libraries using the NEBNext Ultra Directional RNA Library Prep Kit for Illumina according to the manufacturer's recommended protocol. In brief, 1 μg total RNA was poly(A)-selected for each sample, converted to double-stranded cDNA followed by fragmentation and ligation of sequencing adapters. The library was then PCR-amplified for 8 cycles using barcoded PCR primers, purified and size-selected using AMPure XP Beads before loading onto an Illumina NextSeq 2000 for 100 bp single-read sequencing.

**RNA-seq analysis.** Sequenced reads were aligned to the GRCm39 reference genome (Ensembl, v.104; http://uswest.ensembl.org/Mus_musculus/Info/Index), and gene counts were quantified using Salmon (v.1.5.1)[59]. Differential gene expression analysis and *P*-value calculation were performed by DESeq2 (v.1.32.0)[58]. Gene Ontology enrichment analysis was performed using Metascape[59] with the Gene Prioritization by Evidence Counting setting.

### Behavioural and physiological assays
**Mechanical threshold.** Mice were acclimated for 1 h in von Frey chambers. The 50% mechanical threshold was measured with calibrated von Frey filaments (0.07, 0.16, 0.4, 0.6, 1.0, 1.4, 2.0, 4.0 and 6.0 g) using the up–down method[60].

**Two-temperature choice assay.** The two-temperature choice test apparatus was set up as previously described[61]. Lanes were evenly divided between two different temperature plates set at 30 °C and 18 °C, and individual mice were placed in one of the lanes. The mice were given 10 min to acclimatize and were then tracked for 1 h using the EthoVision tracking system (Noldus Information Technology) in a dark room with infrared lighting. The total time spent in each temperature zone was analysed.

**Core body temperature measurement.** Core temperature was measured using a thermocouple rectal probe (World Precision Instruments). Room temperature core body temperature was taken when the mice in thermoneutral condition moved to room temperature for more than 24 h.

**Blood pressure and heart rate measurement.** Blood pressure and heart rate were measured using the tail-cuff method with the CODA High Throughput Non-Invasive Blood Pressure System (Kent Scientific) as previously described[62].

**Targeted detection of norepinephrine in bulk iWAT.** Frozen iWAT tissues were lysed in 4× weight of 0.1 mol l$^{-1}$ perchloric acid, centrifuged and run through a 30 kDa filtration tube (Millipore). The filtrates were analysed on an Agilent 6470 Triple Quadrupole (QQQ) liquid chromatography–mass spectrometry (LC–MS) system using electrospray ionization (ESI) in positive mode. The AJS ESI source parameters were set as follows: the gas temperature was set at 250 °C with a gas flow of 12 l min$^{-1}$ and the nebulizer pressure at 25 p.s.i. The sheath gas temperature was set to 300 °C with the sheath gas flow set at 12 l min$^{-1}$. The capillary voltage was set to 3,500 V. Separation of metabolites was conducted on an Agilent Eclipse Plus C18 LC column (3.5 μm, 4.6 × 100 mm, 959961-902). Mobile phases were as follows: buffer A,

water with 0.1% formic acid; buffer B, acetonitrile with 0.1% formic acid. The LC gradient started at 5% B from 0 to 2 min. The gradient was then increased linearly to 5% A/95% B from 2 to 23 min. From 23 to 28 min, the gradient was maintained at 5% A/95% B; and from 28 to 29 min, the gradient went back to the starting concentration of 5% B. The flow rate was maintained at 0.7 ml min$^{-1}$ throughout the run. Multiple reaction monitoring was performed for noradrenaline, looking at the transition of $m/z = 170.1$ as the precursor ion to the $m/z = 152$ fragment. The dwell time was 200, the fragmentor set at 60, the collision energy set at 4 and the cell accelerator voltage set at 4.

### Metabolic studies
Mice received bilateral sensory ablation were fed a high-fat diet (HFD) (D12492, Research Diets) under thermoneutrality at 30 °C. HFD feeding was started at 1 month after surgery (aged around 11–12 weeks old). Body weight was measured every week. Fasting glucose was taken and glucose tolerance tests were performed at 9 weeks on HFD, and plasma insulin was measured at 13 weeks on HFD.

**Glucose tolerance test.** Mice were fasted for 4 h (08:30–12:30) before intraperitoneal injection of glucose (1 g kg$^{-1}$). Blood glucose levels were measured at the indicated time point using the OneTouch Ultra 2 blood glucose meter.

**Plasma insulin measurement.** Fasting blood samples were retro-orbitally obtained 3 h into the light cycle after fasting for 14 h. Plasma was separated in heparin-treated tubes (Microvette CB 300 LH). Plasma insulin was measured with an ELISA kit (Crystal Chem, 90080) and read using the BioTek Cytation 5 Imaging Reader.

### Western immunoblotting
Whole-tissue protein lysate was extracted in RIPA buffer (G-Biosciences) containing Halt proteinase inhibitor (Thermo Fisher Scientific, 78430) and phosphatase inhibitor (Thermo Fisher Scientific, 78420). Protein lysate was determined using the bicinchoninic acid assay (Pierce). Protein lysates were denatured in Laemmli buffer (BioRad), resolved by 4–12% Mini-PROTEAN TGX SDS–PAGE (BioRad) and transferred to a polyvinylidene difluoride membrane. The membrane was incubated with primary antibodies diluted in EveryBlot blocking buffer (Bio-Rad) overnight at 4 °C and then incubated with secondary antibody anti-rabbit HRP (Jackson Immuno Research, 711-036-152, 1:10,000) or anti-mouse HRP (Jackson Immuno Research, 715-036-150, 1:10,000) diluted in EveryBlot at room temperature. The results were visualized using SuperSignal West Pico PLUS Chemiluminescent Substrate (Invitrogen). The following antibodies were used for immunoblotting: anti-p-HSL(Ser660) (Cell Signaling, 45804, 1:1,000), anti-HSL (Cell Signaling, 4107, 1:1,000), anti-α-tubulin (Abcam, 7291, 1:10,000). Specifically, HSL was blotted after stripping the p-HSL membrane.

### Imaging analysis and quantification
All of the images were analysed using ImageJ. 3D volume images were rendered in Imaris.

**Quantification of TH$^+$ DRG nerves in the iWAT.** Regions of 40 μm × 40 μm × 40 μm ($x,y,z$) were randomly selected and maximally projected over $z$ using customized ImageJ scripts in the whole stacks of intraganglionically labelled iWAT with TH staining. Only regions containing positive TdTomato (DRG) signals were retained. The thickness of TdTomato-positive fibres was measured using the ImageJ straight line function across two different places in the view and averaged. Fibres with widths of less than 2.5 μm (arbitrary cut-off) were considered to be thin fibres. If the TH-647 channel showed overlap with the TdTomato signal, the view was considered to be positive. We quantified 13 images from 3 biological replicates, and a total of 77 views containing the TdTomato positive thin fibres were quantified for TH positivity.

**Quantification of nerve density in flank skin and iWAT.** Regions of 80 μm × 80 μm × 20 μm ($x,y,z$) were randomly selected and maximally projected over $z$ using customized ImageJ scripts in the whole stacks of flank skin or iWAT from mice received intraganglionic labelling. Areas containing nerve fibres were automatically segmented using auto thresholding in ImageJ. Nerve density was calculated as $Area_{Nerve}/Area_{Total}$. Only views containing nerve signals were retained for quantification. We quantified 466 views from 39 images from 3 biological replicates for flank skin, and 468 views from 31 images from 2 biological replicates for iWAT.

**Quantification of intraepidermal nerve fibres.** Quantification of intraepidermal nerve fibres with antibodies against βIII-tubulin was determined according to the number of nerve fibres crossing the basement membrane in the cross-sections of flank skin. Scoring was performed in a blinded manner on all of the images and post hoc registered to the condition. Ipsilateral and contralateral flank skin from 4 mice (10–15 sections per tissue) was quantified.

**Quantification of ROOT/AAV9 labelling in ganglia.** DRGs and SChGs were taken from mice injected with AAV (ROOT or AAV9) and CTB for whole-mount imaging as described above. T11–L3 DRGs and T12 SChGs were quantified for cell numbers labelled by AAV and CTB. The percentage of AAV and CTB double-positive cells in all AAV labelled cells were quantified in T13 and L1 DRGs as intraganglionic surgery was performed on T13 and L1 DRGs for later in vivo experiments.

**Quantification of CTB/ROOT/AAV9-labelled DRG soma sizes.** Soma sizes were quantified manually in full stack images from DRG whole-mount imaging using Fiji. We noticed that injecting CTB conjugated to different fluorophores (that is, CTB-488 versus CTB-647) into the iWAT may result in a slight difference in the distribution of the soma diameter of labelled DRG neurons; we therefore exclusively used CTB-647 when quantifying CTB-labelled DRG soma sizes.

**Quantification of liver fluorescence intensity.** For liver fluorescence determination in the characterization of ROOT, the same views were acquired with two different exposure times. The images with a longer exposure time were used to determine the shape of the tissue, and the images with a shorter exposure time were used to quantify intensity to avoid over-saturation. The region of interest of the liver and the background were drawn manually in ImageJ, and the liver fluorescence intensity was defined as Mean Intensity$_{liver}$ − Mean Intensity$_{background}$.

## Study design

No statistical methods were used to calculate the sample size. The sample size was determined on the basis of previous studies and literature in the field using similar experimental paradigms. No data were excluded, except for mice with deteriorating health issues after surgery or during the experiment and mice with missed viral targeting assessed by histology. Data were collected in a blinded manner and post hoc registered to the condition and analysed accordingly to prevent any bias.

## Statistics and reproducibility

All non-RNA-seq analysis was performed using GraphPad Prism 9 (v.9.3.1) with the indicated statistical tests. Paired two-tailed Student's $t$-tests were performed to compare one animal's ipsilateral and contralateral sides. The individual animal's left or right sides were randomly assigned for unilateral treatment. Mice were randomly assigned for bilateral treatment. Sample sizes for each experiment are reported in the figure legends. All in vivo experiments were performed at least twice or grouped from two independent cohorts with the same conclusion, except HFD treatment was performed in one cohort. For representative images, the numbers of experimental repetitions were as follows: Figs. 1b–d (four), 1e,g,h,j (three), 2a–b,d (two), 4a (four), 4c,d (two) and Extended Data Figs. 1a (three), 1b (four), 1c (two), 2a (six), 2f (three), 2h–i,k (two), 3a,d,e (three), 3b (two), 4a,h (two); 5b (two), 7d (three), 7e (two).

## Reporting summary

Further information on research design is available in the Nature Research Reporting Summary linked to this article.

## Data availability

Bulk RNA-seq data have been deposited at the Gene Expression Omnibus under accession number GSE207664. All the numeric data in this study are included in the Supplementary Information. All other data supporting the findings of this study are too large for public deposit and are available from the corresponding authors. Source data are provided with this paper.

## Code availability

The code used for sequencing analysis and FIJI analysis is deposited at GitHub (https://github.com/yelabscripps).

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

**Acknowledgements** We thank E. Saez, P. Cohen, C.-H. Lee, K. L. Marshall and I. Daou for their input; B. E. Deverman for advice with AAV vector engineering; K. Deisseroth, D. Gibbs and C. Ramakrishnan for the mScarlet and sfGFP plasmids; J. Stirman and K. Spencer for the imaging support; the staff at Scripps genomics core and Sanford Burnham Prebys histology core for sample preparation; A. Chesler and R. Z. Hill for their feedback on manuscript; and all members of the Ye laboratory, Patapoutian laboratory and Dorris Neuroscience Center for their support and feedback. This work was supported by the Howard Hughes Medical Institute; NIH grants R35 NS105067 and R01AT012051 (to A.P.); NIH Director's New Innovator Award DP2DK128800 (to L.Y.), NIDDK K01DK114165 (to L.Y.), Whitehall Foundation (to L.Y.) and Baxter Foundation (to L.Y.); Y.W. was supported by the Dorris Scholars fellowship. Y.Z. is a Merck Fellow of the Damon Runyon Cancer Research Foundation, DRG-2405-20. M.D.M.-G. was supported by the Fundacion Alfonso Martin Escudero postdoctoral fellowship.

**Author contributions** Y.W., A.P. and L.Y. conceived and designed the study. Y.W., V.H.L., Y.Z., V.S.N., M.L. and M.R.S.-V. performed the experiments and analysed the data. D.Y. and K.W. contributed to in vivo animal experiments. M.D.M.-G., V.L.L. and J.Z.L. performed noradrenaline measurements. Y.W., A.P. and L.Y. wrote the manuscript with input from all of the authors. All of the authors provided input and reviewed the manuscript.

**Competing interests** The authors declare no competing interests.

**Additional information**
**Correspondence and requests for materials** should be addressed to Ardem Patapoutian or Li Ye.

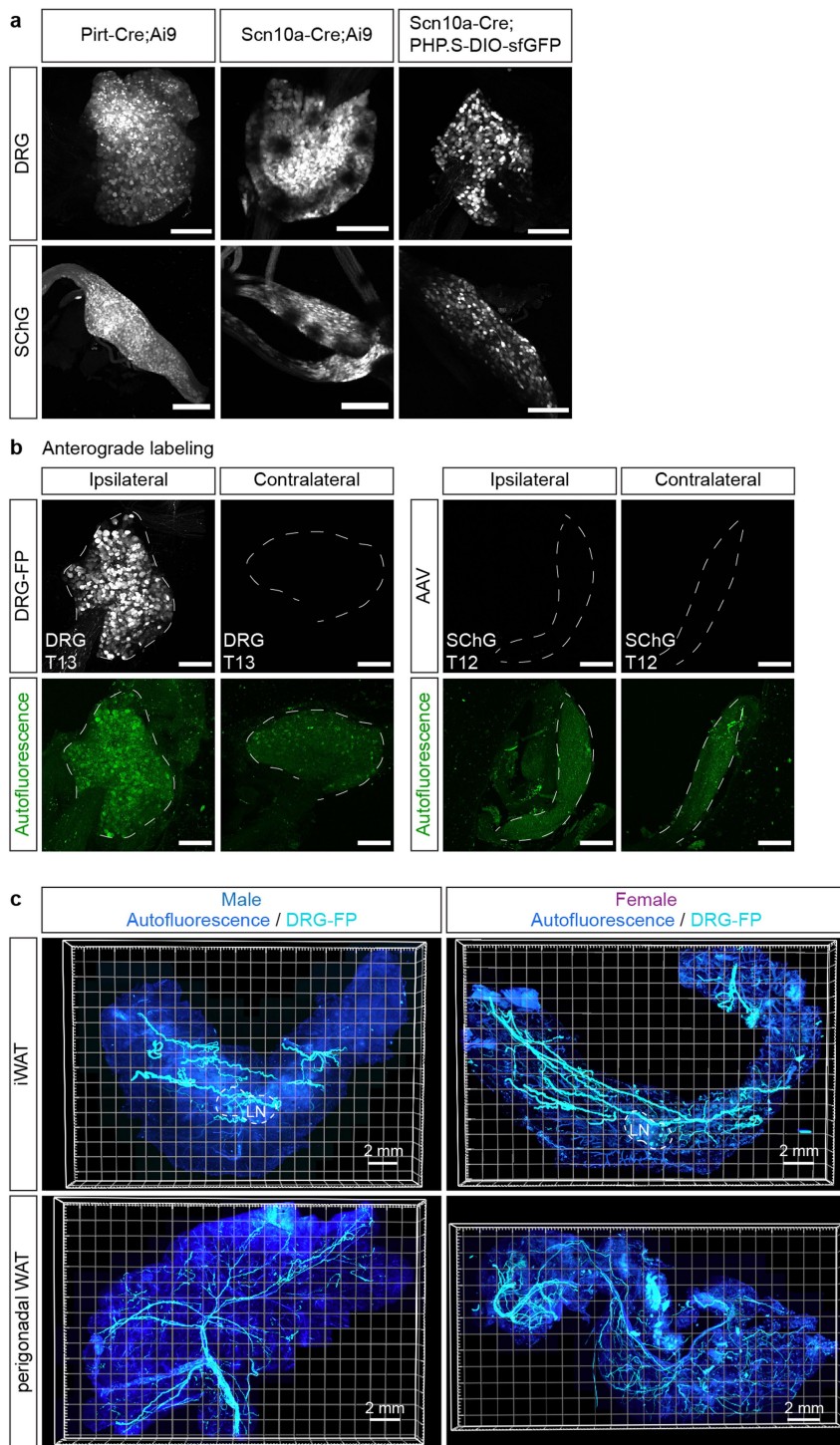

**Extended Data Fig. 1 | Anterograde labelling maps somatosensory innervation in adipose tissues. a**, Representative images of DRG (T13), sympathetic chain ganglia SChG (T12) from Pirt-Cre;Ai9, Scn10a-Cre;Ai9 and Scn10a-Cre with systemic viral labelling (PHP.S-DIO-sfGFP) (3 mice per line). Scale bar: 200 μm. **b**, Representative images of DRG (T13), SChG (T12) from mice with unilateral intraganglionic viral injection in T13 DRG (from 3 mice). Autofluorescence (647 nm laser) is used to show the tissue outline. Scale bar: 200 μm. **c**, Representative images of iWAT and perigonadal WAT (pgWAT, refers to eWAT for male and periovarian WAT for female) from mice with intraganglionic injections of AAV expressing fluorescent protein (FP) in T13 and L1 DRGs (from 3 mice).

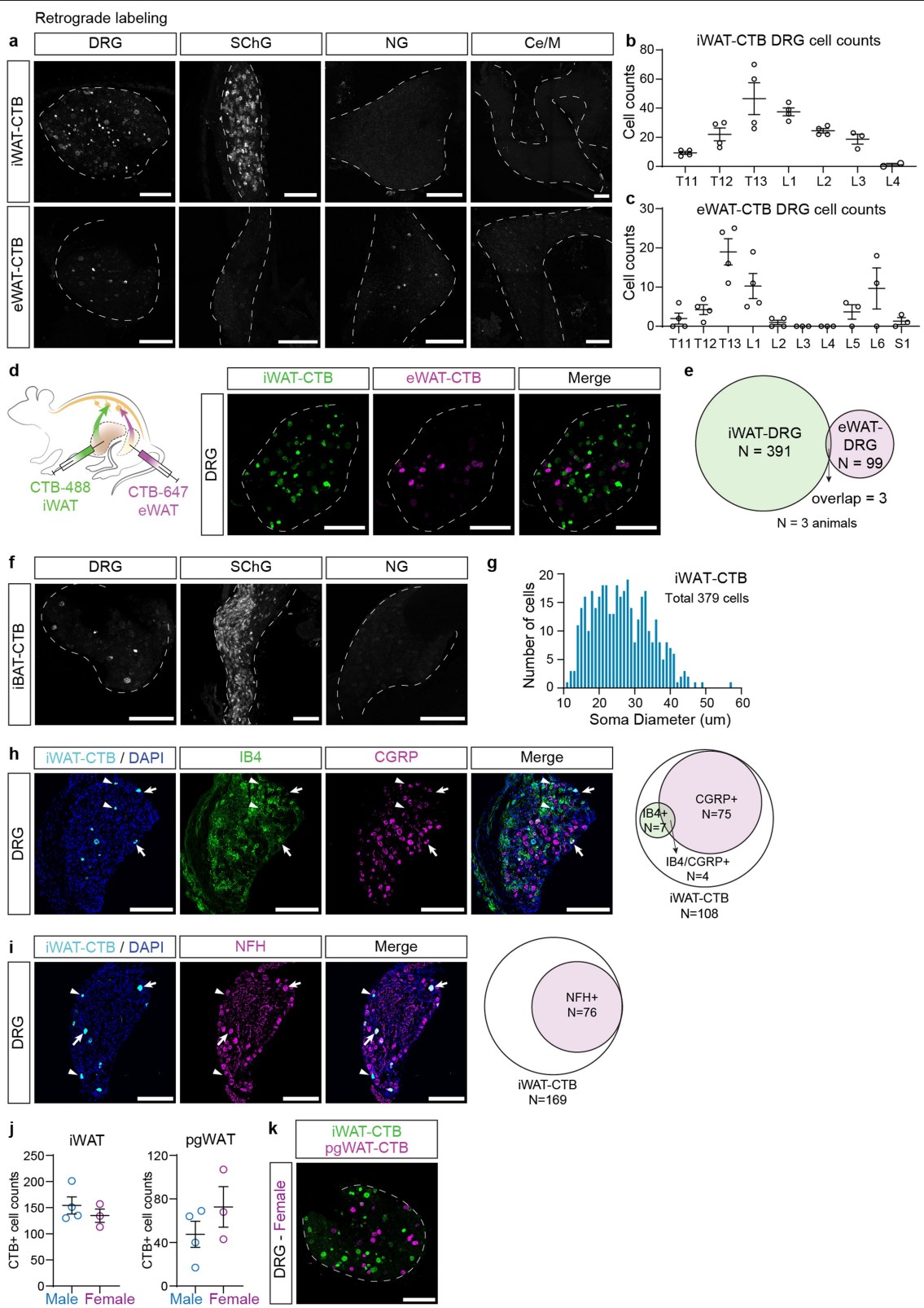

**Extended Data Fig. 2 | Retrograde labelling confirms somatosensory innervation of adipose tissues. a**, Representative images of DRG (T13), SChG (T12), nodose ganglion (NG), and celiac/superior mesenteric complex (Ce/M) from mice with CTB-647 injected into iWAT or eWAT. **b–c**, Quantification of CTB labelled cell numbers in DRG labelled from iWAT (**b**) or eWAT (**c**) along the vertebral levels. n = 4 per group. **d**, Schematics of dual-colour CTB labelling from iWAT and eWAT (left) and representative images of T13 DRG. **e**, Quantification of CTB positive cells from iWAT and eWAT. (n = 3 mice). **f**, Representative images of DRG (T3), SChG (stellate ganglion and T1 SChG) and NG from mice with CTB-488 injected into iBAT. **g**, Quantification of CTB labelled DRG soma size distribution from mice with CTB-647 injected into iWAT (n = 3 mice). **h–i**, Representative images and quantification of cell type of CTB labelled DRG neurons (from 4 mice). **j**, Quantification of total CTB labelled neurons from DRG (T11-L6) in male and female mice with CTB injected into iWAT or eWAT (n = 4 for male mice, and n = 3 for female mice). **k**, Representative image of DRG (L1) from female mice with dual-colour CTB labelling from iWAT and pgWAT (from 3 mice). All values are mean ± s.e.m. in **b**, **c**, **j**. Scale bar: 200 μm.

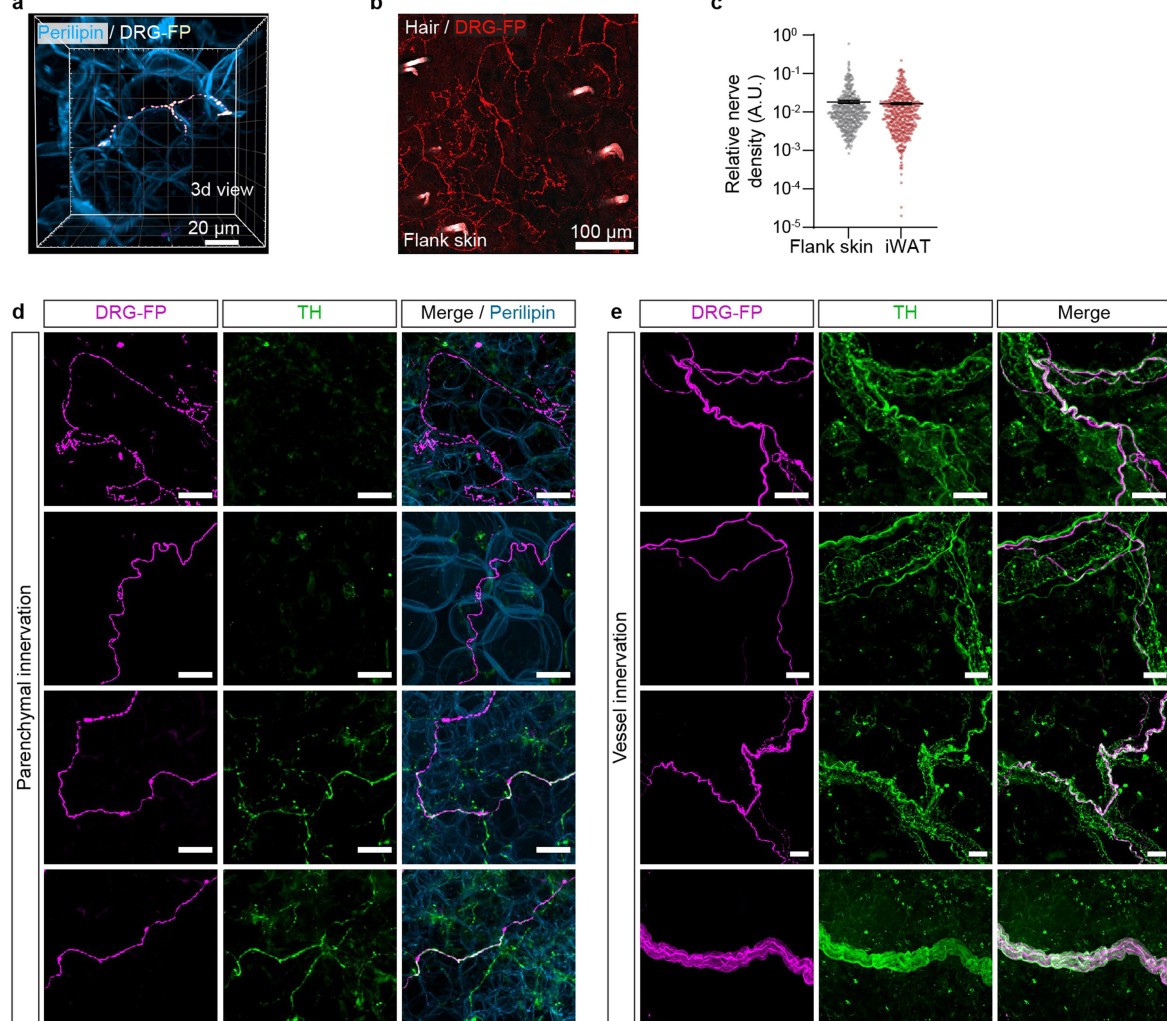

**Extended Data Fig. 3 | Anterograde labelling reveals the morphological features of somatosensory fibres in iWAT. a**, 3D view of intraganglionically fluorescent protein (FP) labelled DRG (T13 and L1) fibres in close apposition to adipocytes. **b**, Representative image of intraganglionically FP labelled DRG (T13 and L1) fibres in flank skin. **c**, Quantification of relative nerve density of intraganglionically FP labelled DRG (T13 and L1) fibres in flank skin (466 views from 39 images from 3 biological replicates) and iWAT (468 views from 31 images from 2 biological replicates). All values are mean ± s.e.m. **d**–**e**, Representative images of intraganglionically FP labelled DRG (T13 and L1) fibres in adipose parenchyma (**d**) and travelling along the vessel (**e**). Scale bar: 30 µm.

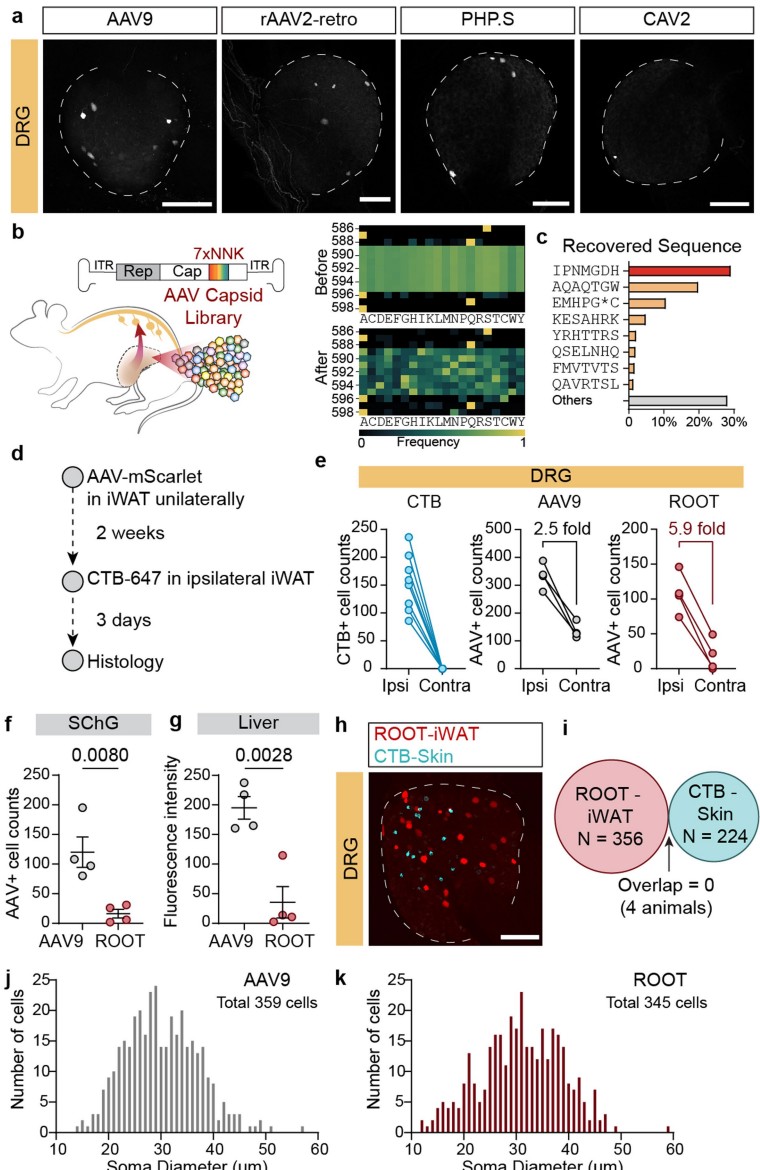

**Extended Data Fig. 4 | Development and characterization of retrograde vector optimized for organ tracing (ROOT). a**, Representative images of DRG (T13) from mice with AAV9, rAAV2-retro, PHP.S, or CAV2 injected into iWAT. Scale bar: 200 µm. **b-c**, Development of ROOT. **b**, Schematics of *in vivo* selection of retrograde vector and amino acid enrichment after one-round selection. **c**, Quantification of the abundance of recovered sequence with peptide insertion. **d**–**g**, Comparison of ROOT and AAV9. **d**, Workflow of ROOT and AAV9 comparison. **e**, Quantification of AAV+ cell numbers in ipsilateral and contralateral DRGs (T11-L3). **f**, Quantification of AAV+ cell numbers in ipsilateral SChG (T12). **g**, Quantification of bulk fluorescence intensity of liver. All values are mean ± s.e.m. Statistics determined by two-tailed unpaired t test in **f**, **g**. **h**–**i**, Representative image of DRG (T13) (**h**) and quantification of cell numbers (**i**) from mice with ROOT-mScarlet injected in iWAT and CTB injected in flank skin. Scale bar: 200 µm. **j**–**k**, Quantification of AAV labelled DRG soma size distribution from mice with AAV9 (**j**) or ROOT (**k**) injected into iWAT (n = 2 mice for AAV9, n = 5 mice for ROOT).

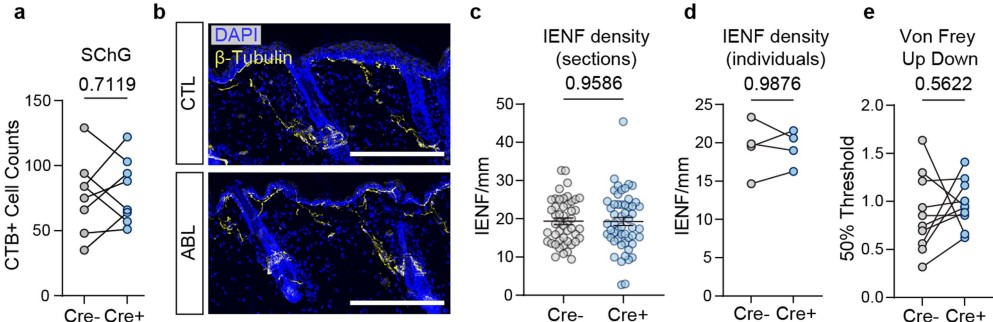

**Extended Data Fig. 5 | Characteriztaion of Cre-dependent unilateral ablation of iWAT-DRGs. a**, Quantification of CTB labelled cells in SChG (T12). n = 8. Statistics determined by two-tailed paired t test. **b**, Representative images of flank skin nerve fibres. Scale bar: 200 μm. **c**–**d**, Quantification of intra epidermal nerve fibre (IENF) density of flank skin. n = 4 mice, 10-15 non-continuous sections were quantified per sample. **c**, Pooled IENF density of flank skin from Cre- side (n = 48) and Cre+ side (n = 53). All values are mean ± s.e.m. Statistics determined by two-tailed unpaired t test. **d**, Average IENF density per animal. Statistics determined by two-tailed paired t test. **e**, Mechanical threshold of hindpaws from mice with unilateral sensory ablation of iWAT innervation. n = 10 mice. Statistics determined by two-tailed paired t test.

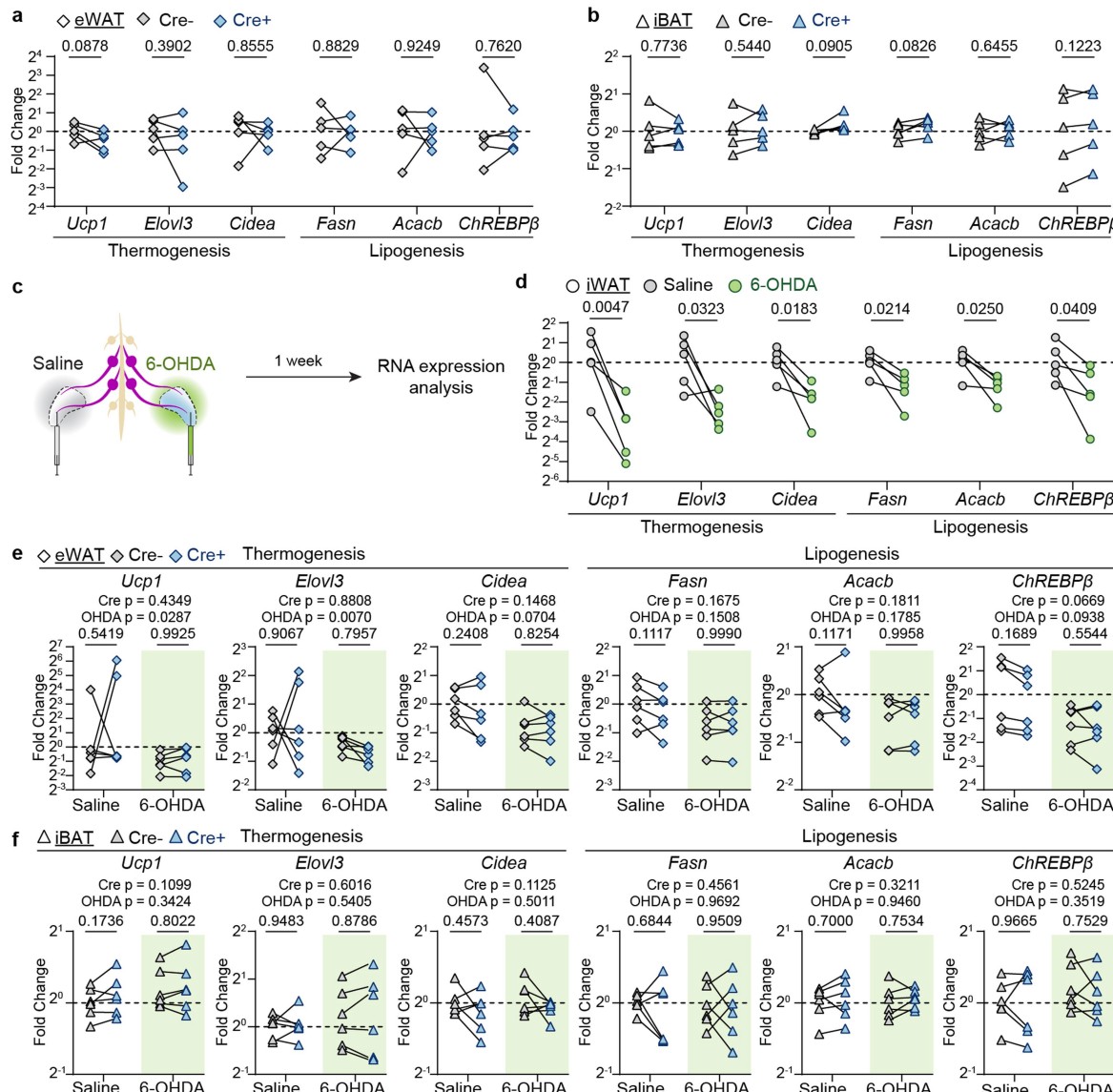

**Extended Data Fig. 6 | Gene expression analysis of Cre-dependent unilateral ablation of iWAT-DRGs. a–b,** Quantitative RT-PCR analysis of eWAT (**a**) and iBAT (**b**) after Cre-dependent unilateral sensory ablation in iWAT. n = 5 mice. Statistics determined by two-tailed paired t test. **c–d,** Chemical denervation of sympathetic innervation of iWAT. **c,** Workflow of sympathetic chemical denervation. 6-OHDA was injected into iWAT unilaterally.

**d,** Quantitative RT-PCR analysis of iWAT after sympathetic chemical denervation. Statistics determined by two-tailed paired t test. **e–f,** Quantitative RT-PCR analysis of eWAT (**e**) and iBAT (**f**) after Cre-dependent unilateral sensory ablation and bilateral sympathetic chemical denervation. n = 6 mice per group. Statistics determined by 2-way ANOVA.

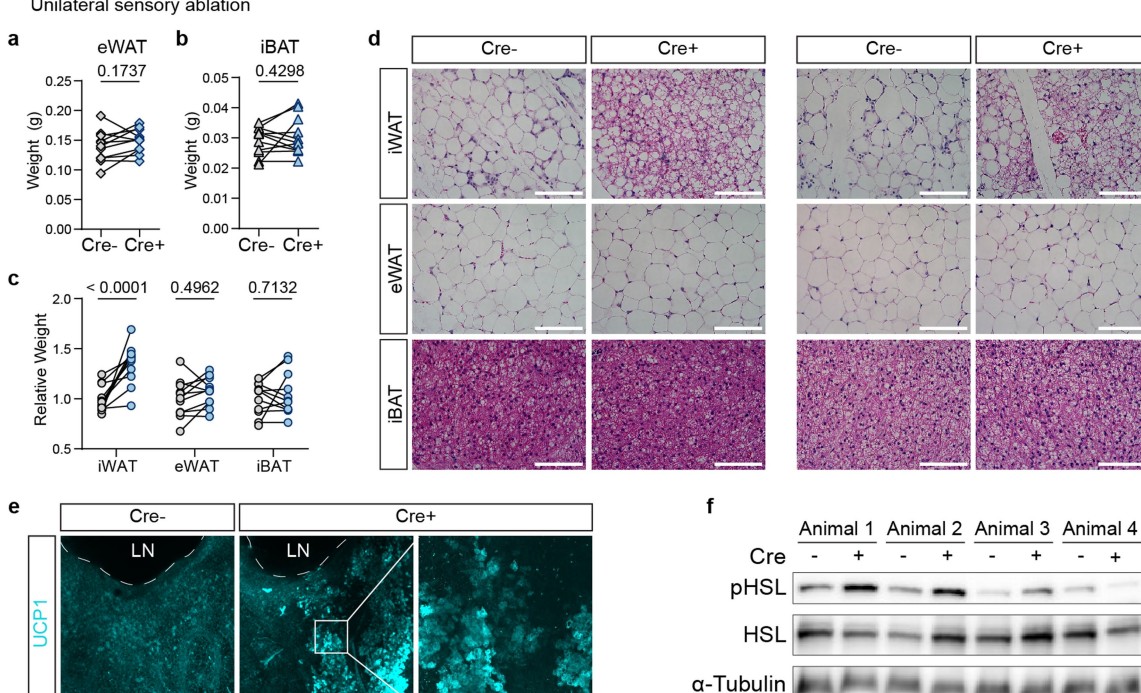

**Extended Data Fig. 7 | Characterization of morphological changes of iWAT after sensory ablation. a–b**, Fat mass of eWAT (**a**) and iBAT (**b**) after unilateral sensory ablation of iWAT. n = 11 mice. Statistics determined by two-tailed paired t test. **c**, Normalized weight of iWAT, eWAT and iBAT. n = 11 mice. Statistics determined by 2-way ANOVA. **d**, Representative histological images of iWAT, eWAT, iBAT from mice (n = 3) with unilateral sensory ablation of iWAT. Scale bar: 100 μm. **e**, Digital slice views (200 μm) of UCP1 staining in iWAT from mice (n = 3) with unilateral sensory ablation, showing lymph node (LN) as a landmark. Scale bar: 500 μm. **f**, Western blot of p-HSL, HSL and α-Tub in iWAT from mice (n = 4) with unilateral sensory ablation.

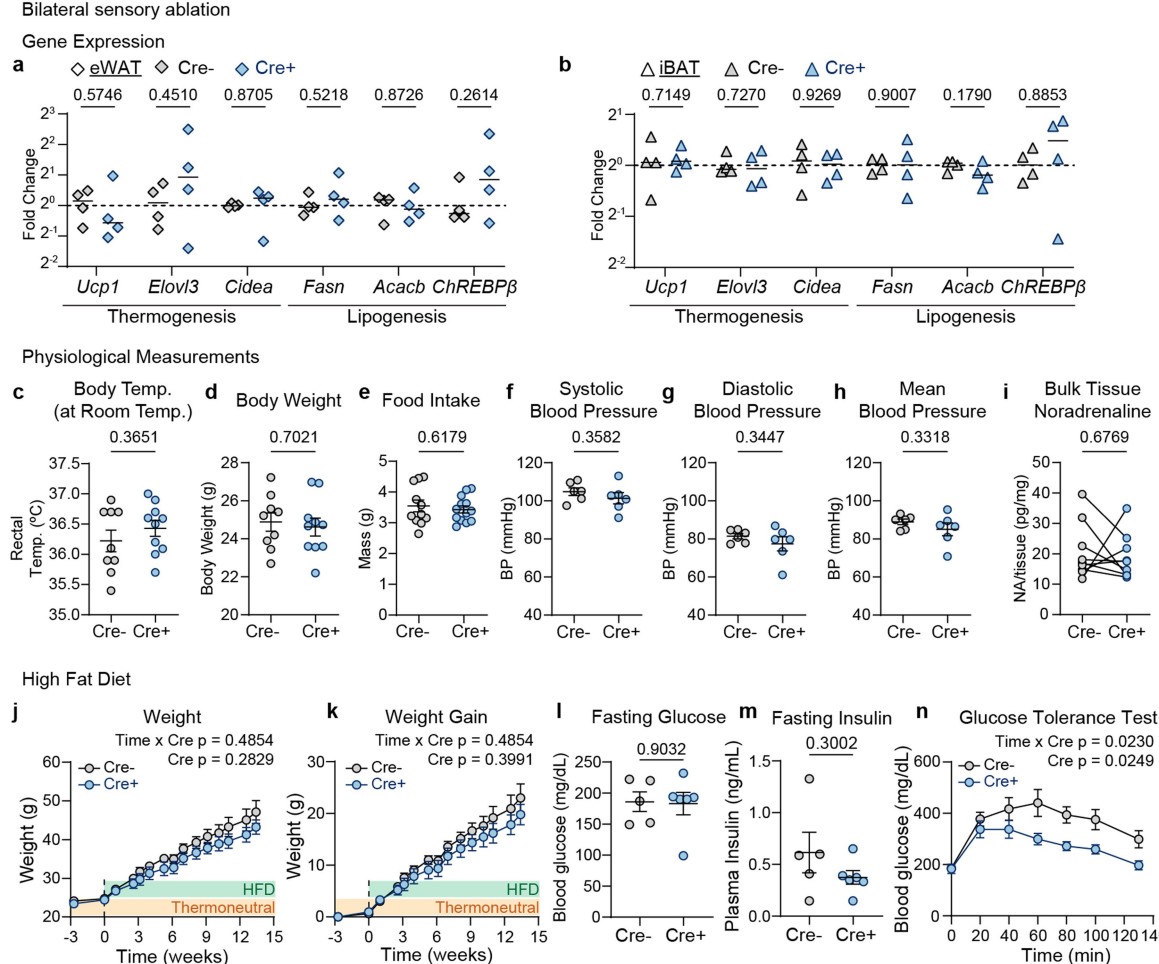

**Extended Data Fig. 8 | Characterization of physiological changes of iWAT after sensory ablation. a–b**, Quantitative RT-PCR analysis of eWAT (**a**) and iBAT (**b**) from mice with bilateral sensory ablation of iWAT (n = 4 mice per group). **c–h**, Physiological measurement of mice with bilateral sensory ablation. **c**, Rectal temperature at room temperature of Cre- (n = 9) and Cre+ (n = 10). **d**, Body weight of Cre- (n = 9) and Cre+ (n = 10). **e**, 24 h food intake of Cre- (n = 11) and Cre+ (n = 13). **f–h**, Systolic (**f**), diastolic (**g**) and mean (**h**) blood pressure of Cre- (n = 6) and Cre+ (n = 6). **i**, Bulk iWAT noradrenaline (NA) amount in mice with unilateral sensory ablation (n = 9). **j–n**, Metabolic measurement of

mice with bilateral sensory ablation on high-fat diet (HFD) in thermoneutral temperature. **j–k**, Body weight (j) and body weight gain (k) of Cre- (n = 5) and Cre+ (n = 6) mice on HFD. **l**, Fasting glucose levels in Cre- (n = 5) and Cre+ (n = 6) mice after 9 weeks of HFD. **m**, Fasting plasma insulin levels in Cre- (n = 5) and Cre+ (n = 6) mice after 14 weeks of HFD. **n**, IP-glucose tolerance test (1 g/kg) in Cre- (n = 5) and Cre+ (n = 6) mice after 9 weeks of HFD. **c–h, j–n** are shown as mean ± s.e.m. Statistics determined by two-tailed unpaired t test in **a–b**; two-tailed unpaired t test with Welch's correction in **c–h, l–m**; 2-way ANOVA in **j, k, n**.

|---|---|

# Reporting Summary

## Statistics

For all statistical analyses, confirm that the following items are present in the figure legend, table legend, main text, or Methods section.

| n/a | Confirmed | |
|---|---|---|
| ☐ | ☒ | The exact sample size (*n*) for each experimental group/condition, given as a discrete number and unit of measurement |
| ☐ | ☒ | A statement on whether measurements were taken from distinct samples or whether the same sample was measured repeatedly |
| ☐ | ☒ | The statistical test(s) used AND whether they are one- or two-sided *Only common tests should be described solely by name; describe more complex techniques in the Methods section.* |
| ☐ | ☒ | A description of all covariates tested |
| ☐ | ☒ | A description of any assumptions or corrections, such as tests of normality and adjustment for multiple comparisons |
| ☐ | ☒ | A full description of the statistical parameters including central tendency (e.g. means) or other basic estimates (e.g. regression coefficient) AND variation (e.g. standard deviation) or associated estimates of uncertainty (e.g. confidence intervals) |
| ☐ | ☒ | For null hypothesis testing, the test statistic (e.g. *F*, *t*, *r*) with confidence intervals, effect sizes, degrees of freedom and *P* value noted *Give P values as exact values whenever suitable.* |
| ☒ | ☐ | For Bayesian analysis, information on the choice of priors and Markov chain Monte Carlo settings |
| ☐ | ☒ | For hierarchical and complex designs, identification of the appropriate level for tests and full reporting of outcomes |
| ☒ | ☐ | Estimates of effect sizes (e.g. Cohen's *d*, Pearson's *r*), indicating how they were calculated |

*Our web collection on statistics for biologists contains articles on many of the points above.*

## Software and code

Policy information about availability of computer code

| Data collection | FLUOVIEW 2.4.1.198 (OLYMPUS FV3000RS microscope), SmartSPIM GUI 2.1 (LifeCanvas Technnologies microscope), SmartSPIM Destriping and Stitching (2.0, LifeCanvas Technologies), SpinView (v2.5.0.80, FLIR BFS-U3 camera), BZ-X Viewer (Keyence BZ-X710 microscope), Bio-Rad CFX Manager (3.1, Biorad CFX384 qPCR machine) |
|---|---|
| Data analysis | samtools (v1.10), Salmon (v1.5.1), DESeq2 (v1.32.0), Metascape (v3.5), Fiji (v2.3.0; image projection and video making), IMARIS (v9.2.1; three-dimensional image rendering), Graphpad Prism (v9.3.1) Custom codes used for sequencing analysis and FIJI analysis is deposited to https://github.com/yelabscripps. |

For manuscripts utilizing custom algorithms or software that are central to the research but not yet described in published literature, software must be made available to editors and reviewers. We strongly encourage code deposition in a community repository (e.g. GitHub). See the Nature Portfolio guidelines for submitting code & software for further information.

## Data

Policy information about availability of data

All manuscripts must include a data availability statement. This statement should provide the following information, where applicable:
- Accession codes, unique identifiers, or web links for publicly available datasets
- A description of any restrictions on data availability
- For clinical datasets or third party data, please ensure that the statement adheres to our policy

Bulk RNA-seq data is deposited under accession number GSE207664 (https://www.ncbi.nlm.nih.gov/geo/query/acc.cgi?acc=gse207664). The mice genome dataset (GRCm39 reference genome (Ensembl version 104) used for sequencing alignment can be accessed at http://uswest.ensembl.org/Mus_musculus/Info/Index.

# Field-specific reporting

Please select the one below that is the best fit for your research. If you are not sure, read the appropriate sections before making your selection.

☒ Life sciences ☐ Behavioural & social sciences ☐ Ecological, evolutionary & environmental sciences

For a reference copy of the document with all sections, see nature.com/documents/nr-reporting-summary-flat.pdf

# Life sciences study design

All studies must disclose on these points even when the disclosure is negative.

| | |
|---|---|
| Sample size | No statistical methods were used to calculate the same size. The sample size was determined based on previous studies and literature in the field using similar experimental paradigms (Marshall KL et al. Nature, 2020; Lehnert BP et al. Cell, 2021; Bai L et al. Cell, 2019). |
| Data exclusions | No data were excluded, except mice with deteriorating health issues after surgery or during the experiment and mice with missed viral targeting assessed by histology. |
| Replication | All experiments were repeated at least twice with the same conclusions. All in vivo experiments were performed at least twice or grouped from two independent cohorts with the same conclusion, except long term high fat diet treatment was performed in one cohort for timing consideration. The exact number of replicated experiments are provided in the Statistics and reproducibility section. |
| Randomization | The individual animal's left or right sides were randomly assigned for unilateral treatment. Mice were randomly assigned for bilateral treatment. |
| Blinding | Data were collected blind, and post hoc registered to the treatment conditions and analyzed to prevent any bias. |

# Reporting for specific materials, systems and methods

We require information from authors about some types of materials, experimental systems and methods used in many studies. Here, indicate whether each material, system or method listed is relevant to your study. If you are not sure if a list item applies to your research, read the appropriate section before selecting a response.

## Materials & experimental systems

| n/a | Involved in the study |
|---|---|
| ☐ | ☒ Antibodies |
| ☐ | ☒ Eukaryotic cell lines |
| ☒ | ☐ Palaeontology and archaeology |
| ☐ | ☒ Animals and other organisms |
| ☒ | ☐ Human research participants |
| ☒ | ☐ Clinical data |
| ☒ | ☐ Dual use research of concern |

## Methods

| n/a | Involved in the study |
|---|---|
| ☒ | ☐ ChIP-seq |
| ☒ | ☐ Flow cytometry |
| ☒ | ☐ MRI-based neuroimaging |

## Antibodies

| | |
|---|---|
| Antibodies used | Anti-Perilipin-1 (Cell Signaling #9349, 1:400); Anti-TH-647 (BioLegend #818008, 1:300); Anti-beta-Tubulin (Abcam ab18207, 1:1000); Anti-Rabbit-488 (Jackson Immuno Research 711-546-152, 1;400); Anti-Rabbit-647 (Jackson Immuno Research 711-606-152, 1:400); Anti-p-HSL (S660) (Cell Signaling #45804, 1:1000), Anti-HSL (Cell Signaling #4107, 1:1000), Anti-α-Tubulin (Abcam #DM1A, 1:10000). Anti-Ucp1 (Abcam #ab10983, 1:200). Anti-rabbit HRP (Jackson Immuno Research 711-036-152, 1:10000), Anti-mouse HRP (Jackson Immuno Research 715-036-150, 1:10000),  Anti-CGRP (Immunostar 24112, 1:1000). |
| Validation | Anti-Perilipin-1 (Cell Signaling #9349): mouse tissues, Maryanovich M., et al. Nat Med. (2018); Li C., et al. Nat Commun. (2020)<br>Anti-TH-647 (BioLegend, #818008): Non-conjugated form (clone 2/40/15, BioLegend #818001) validated on mouse tissues in: Ku T. et al, Nat Methods. (2020); Ku T. et al. Nat Biotechnol (2016).<br>Anti-beta-Tubulin (Abcam ab18207): mouse tissues, Walsh CM, et al. Elife (2019). Latremoliere A, et al. Cell Rep. (2018).<br>Anti-Ucp1 (Abcam #ab10983): mouse tissues, Chi J et al. Elife (2021).<br>Anti-p-HSL (S660) (Cell Signaling #45804): mouse tissues, Ding L et all, Nat Metab (2021).<br>Anti-HSL (Cell Signaling #4107): mouse tissues, Ding L et all, Nat Metab (2021).<br>Anti-α-Tubulin [DM1A] (Abcam #ab7291): HEK cell lines, Khan OM et al, Nat Commun (2021).<br>Anti-CGRP (Immunostar 24112): mouse tissues, Hill RZ et al. Nature. (2022). |

# Eukaryotic cell lines

| | |
|---|---|
| Cell line source(s) | HEK293FT (Invitrogen R70007) |
| Authentication | HEK293FT was purchased from Invitrogen, no further authentication of line identity was performed. |
| Mycoplasma contamination | HEK293FT was purchased from Invitrogen, no further test for mycoplasma contamination was performed. |
| Commonly misidentified lines (See ICLAC register) | No commonly misidentified lines have been used. |

# Animals and other organisms

| | |
|---|---|
| Laboratory animals | Mice were group-housed in standard housing with 12:12h light:dark with ad libitum access to chow diet and water, with room temperature kept around 22C and humidity kept between 30-80% (not controlled), except for food intake measurement (single housed) and thermoneutral exposure experiments (30C). Mice of at least 6 weeks from the following strains were used for this study wild-type (WT) C57BL/6J (Jackson stock #000664), B6.Cg-Gt(ROSA)26Sortm9(CAG-tdTomato)Hze/J (Jackson stock #007909, Ai9), Pirt-cre (Kim, A. Y. et al. Cell 2008) , Scn10a-cre (Agarwal, N. et al. Genesis 2004). Both genders were used for anatomical mapping studies, while male mice were used for in vivo functional experiments. |
| Wild animals | The current study did not utilize wild animals. |
| Field-collected samples | The current study did not utilize field-collected samples. |
| Ethics oversight | All experimental protocols were approved by The Scripps Research Institute Institutional Animal Care and Use Committee (Animal protocol 18-0001, 08-0136) and were in accordance with the guidelines from the NIH. |

Note that full information on the approval of the study protocol must also be provided in the manuscript.

