## [Peer Review File · Nature]

Manuscript Title: The role of somatosensory innervation of adipose tissues

Reviewer Comments & Author Rebuttals

Reviewer Reports on the Initial Version:

Referees' comments:

Referee #1 (Remarks to the Author):

Wang et al. examined the role of somatosensory innervation of adipose tissues. Using a novel retrograde tracing method, ROOT (retrograde vector optimized for organ tracing), in combination of HYBRiD, which allows for direct visualization of the entire projection from T13-DRG soma to the target (inguinal white adipose tissue, iWAT), the authors demonstrated specific innervation to mouse iWAT. Furthermore, their approach allows for selective ablation of sensory neurons innervating iWAT. The authors showed that selective sensory ablation in adipose tissue enhanced the transcription of lipogenic and thermogenetic genes, leading to increases in fat pad, number of beige adipocytes, and body temperature under thermoneutral conditions. They also showed these effects of the sensory neuron ablation can be reversed by sympathetic blockade. The findings are highly significant and the approaches are novel. It is also important to show that DRG sensory neurons act as an inhibitory brake on the local sympathetic function. The following issues should be addressed to improve the manuscript.

1. As the authors pointed out, there are two outstanding questions: what types of sensory neurons innervate fat, and what are they sensing? It will strengthen the paper to address these issues with some experimental efforts.

1a. Please conduct additional characterization of iWAT-innervating neurons simply by double staining and cell size measurement. Are these C-fibers/A-fibers or TH+ only?

1b. I wonder if the ablated neurons are important for thermal sensation. Loss of certain TRP channels in these sensory neurons may be responsible for thermal dysregulation. I understand mechanical sensitivity was tested and these animals were housed at murine thermoneutrality (~29–30C). Is it possible to measure thermal sensitivity change in these mice with ablation?

2. It is suggested that beige fat-innervating sensory neurons modulate adipocyte function by acting as a brake on the sympathetic system. It is unclear how these two systems interact. Any speculation of some sensory mediator that can inhibit sympathetic system function?

3. Figure 1: Please try to quantify the density of nerve innervation in adipose tissue. Is this innervation comparable to skin or muscle innervation?

4. Fig. 1e: CTB-488-labeled neurons look smaller than CTB-647-labeled neurons.

5. Please analyze diameters/sizes of AAV9 infected neurons. Is it possible that AAV9 preferentially labels certain population of sensory neurons?

6. I wonder if you saw sex differences in somatosensory innervations of adipose tissue (iWAT). Male and female mice have different body weights and fat ratios.

Minor

Fig. 4i: The sample size of Cre- group is small (n=5), compared to other groups (n=8-10).

Line 64: It is stated "pan-DRG Cre transgenic mice (such as Pirt-Cre, Scn10a-Cre) also significantly target sympathetic neurons". What about Advillin-Cre?

Referee #2 (Remarks to the Author):

The sympathetic nervous system is well known to control adipose tissue thermogenesis and lipolysis. The new work by Wang et al. reports that somatosensory innervation to the inguinal adipose tissue regulates its function. The authors first developed new viral and imaging tools to visualize sensory nerves and found that DRG axons from soma projected to the inguinal adipose tissues in mice. Subsequently, the authors demonstrated that selective ablation of sensory nerves led to transcriptional changes in the inguinal adipose tissue, including elevated expression of lipogenic genes and thermogenic genes.

The findings are novel and significant for the following reasons: first, the authors employed new tools to precisely dissect the projection of somatosensory neurons into the inguinal adipose tissues. TH+ nerves in adipose tissues have been viewed as sympathetic nerves, but this work showed that a part of them are somatosensory neurons. Second, the authors provided evidence that somatosensory neurons control the function of adipose tissues in vivo. The latter, however, needs additional evidence - this reviewer's comment focuses on the area.

Comment 1. The authors showed that 6-OHDA treatment blunted the effect of sensory ablation on thermogenic and lipogenic genes. Intriguingly, however, sensory ablation stimulated lipogenic genes rather than the lipolytic pathway, suggesting more complicated regulation than enhanced sympathetic nerves. If sensory ablation caused de-repression of the sympathetic activity, it would have activated lipolysis, rather than lipogenesis. Thus, the authors need to further dissect the underlying mechanism. For instance, the authors may test if norepinephrine levels in WAT or circulation are altered by sensory ablation. Analyzing changes in the cellular composition of WAT (e.g., immune cell infiltration) or activities of tissue-resident macrophages, ILC2, etc. could be another possibility.

Comment 2. Beige cells are often clustered near the nerve terminals, and the distribution varies on

the location of adipose tissues. The authors should provide a comprehensive view of inguinal adipose tissues with a logistic relationship between emerged beige adipocytes and sensory projection.

Comment 3. A key question would be to determine the metabolic significance of this neuronal control at a systemic level. The authors should determine the degree to which sensory ablation in adipose tissue affects whole-body energy expenditure and food intake.

Comment 4. An exciting opportunity for the authors is to see how sensory ablation influences body-weight gain in response to an obesogenic diet, such as a high-fat diet. Given the elevated lipogenic genes and thermogenic genes by sensory ablation, it would be intriguing to see how this leads to changes in body weight and fat mass.

Referee #3 (Remarks to the Author):

In a manuscript entitled: "The role of somatosensory innervation of adipose tissue" Wang et al. propose that beige fat-innervating sensory neurons modulate adipocyte function by acting as a brake on the sympathetic system.

This is an interesting ms exploring somatosensory innervation of adipose tissue. By acting as a break on the sympathetic system, the authors show that this type of innervation is an important regulator of adipose tissue function and cellular composition. Over all, the experiments appear well executed with relevant controls. However, there are still some issues that needs to be addressed.

Comments

With the work of the late Timothy J Bartness in mind, it is perhaps a bit oversimplified to state in the Abstract that: "The canonical view is that circulating hormones secreted by the fat convey metabolic state to the brain, which integrates peripheral information and regulates adipocyte function through noradrenergic sympathetic output." Especially relevant in this context is his work using the viral tract tracer, the H129 strain of the herpes simplex virus-1, he discovered for the first time the sensory pathway by which WAT connects with and informs the central nervous system of events occurring in the fat pad Ref 16 (<https://doi.org/10.1152/ajpregu.90786.2008>). Thus, a link between adipose tissue and the CNS, both efferent and afferent, not dependent on circulating hormones has been reported previously. It would be appropriate to acknowledge this fact a bit more clearly.

The in vivo experiments have been performed using C57BL/J mice, an inbred mice strain prone to develop insulin resistance, obesity, and type 2 diabetes. The authors should make it clear that this is a limitation to their study. We don't know if the findings reported can be generalized or if they are a consequence of this propensity. One way around this problem would be to also perform some of the key experiments using a mice strain less prone to develop these conditions such as A/J.

The authors should in greater detail discuss how central sympathetic out-flow is regulated in

response to somatosensory input from WAT. The rabies or herpes virus mediated methodology of retrograde tracing is trans-synaptic and will aid in identifying regions in the CNS that are responsible for sympathetic out-flow regulation. Have the authors considered to use this methodology to also map CNS regulation of sympathetic out-flow?

How specific is the regulation evoked by somatosensory input from WAT? Are there any systemic consequences of ablating sensory afference from WAT? Changes in systemic blood pressure? Heart rate? Systemic glucose tolerance?

The authors need to investigate possible changes in global gene expression not only in ipsi- and contra-lateral fat pads but also in intraabdominal WAT and interscapular BAT to depict any changes at these sites that might be due to altered sympathetic out-flow in response to ablation of somatosensory afference.

Author Rebuttals to Initial Comments:

Response to Reviewers

We are delighted that all reviewers are excited about our discoveries on somatosensory innervation of adipose tissues, calling it “highly significant” and “novel”. We thank the reviewers and editors for the highly constructive suggestions on more detailed characterization of innervation, testing systemic sympathetic functions, and investigating the effects of metabolic challenges such as high-fat diet. We are happy to report that we were able to complete almost all the requested new experiments and included data in the revision.

Referee #1 (Remarks to the Author):

Wang et al. examined the role of somatosensory innervation of adipose tissues. Using a novel retrograde tracing method, ROOT (retrograde vector optimized for organ tracing), in combination of HYBRiD, which allows for direct visualization of the entire projection from T13-DRG soma to the target (inguinal white adipose tissue, iWAT), the authors demonstrated specific innervation to mouse iWAT. Furthermore, their approach allows for selective ablation of sensory neurons innervating iWAT. The authors showed that selective sensory ablation in adipose tissue enhanced the transcription of lipogenic and thermogenetic genes, leading to increases in fat pad, number of beige adipocytes, and body temperature under thermoneutral conditions. They also showed these effects of the sensory neuron ablation can be reversed by sympathetic blockade. The findings are highly significant and the approaches are novel. It is also important to show that DRG sensory neurons act as an inhibitory break on the local sympathetic function. The following issues should be addressed to improve the manuscript.

We thank this reviewer for the positive comments and detailed suggestions on future improvements.

1. As the authors pointed out, there are two outstanding questions: what types of sensory neurons innervate fat, and what are they sensing? It will strengthen the paper to address these issues with some experimental efforts.

1a. Please conduct additional characterization of iWAT-innervating neurons simply by double staining and cell size measurement. Are these C-fibers/A-fibers or TH+ only?

We agree that further characterization of the cell types of fat innervating neurons would strengthen the paper. As suggested, we now have included cell size measurements of CTB labeled DRGs from iWAT in the revised manuscript (Extended Data Fig. 2g). We have also performed double staining against canonical DRG subtype markers (IB4, CGRP, NFH) in the DRGs from CTB retrogradely labeled mice (Extended Data Fig. 2h, i). Our new data shows that the iWAT-innervating sensory neurons are of mixed cell types, with the majority being peptidergic neurons (CGRP+) and myelinated neurons (NFH+). Thus, the iWAT-innervating sensory neurons have both presumptive A- and C- fibers besides Th+ fibers.

1b. I wonder if the ablated neurons are important for thermal sensation. Loss of certain TRP channels in these sensory neurons may be responsible for thermal dysregulation. I understand mechanical sensitivity was tested and these animals were housed at murine thermoneutrality (~29–30C). Is it possible to measure thermal sensitivity change in these mice with ablation?

We have now conducted a temperature preference assay on mice with bilateral sensory ablation (Fig. 4g) and found no difference in thermal sensitivity was associated with bilateral adipose sensory ablation.

2. It is suggested that beige fat-innervating sensory neurons modulate adipocyte function by acting as a brake on the sympathetic system. It is unclear how these two systems interact. Any speculation of some sensory mediator that can inhibit sympathetic system function?

There have been examples in the literature where the sensory system modulates sympathetic activity^{1,2}. Some of the more recent work³⁻⁵ further suggest sensory regulation of distinct autonomic pathways via different circuits. While these new observations (including ours) are very exciting, they are still in their infancy and will most likely require more future work to establish the exact mechanisms by experiments. Nonetheless, we examined systemic sympathetic activity (Fig. 4h, Extended Data Fig. 8f-h) and bulk adipose tissue norepinephrine level (Extended Data Fig. 8i) and did not observe significant changes upon sensory ablation, suggesting that sensory sympathetic interaction may happen downstream of adrenergic receptor(s), or it may induce heterogeneous temporal or local responses within the tissue (which may not be detectable by bulk tissue extract and HPLC-MS).

As the reviewer suggested, we have now included all these speculations based on literature and new data in the discussion (page 9-10, line 196-210).

3. Figure 1: Please try to quantify the density of nerve innervation in adipose tissue. Is this innervation comparable to skin or muscle innervation?

We have now included relative nerve density of flank skin and iWAT from intraganglionically DRG labeled mice (Extended Data Fig. 3b-c). Since the skin innervation is a thin layer, while the fat is more volumetric, there's no established way to compare their nerve density. We took optical sections (80 * 80 * 20 um, x*y*z) and maximally project over z, and automatically segmented the nerve. We then calculate the ratio between total pixels of nerve and total pixels of the ROIs. With this quantification, the fat innervation is comparable to skin innervation. The detailed steps are now included in the methods (page 35-36, line 278-285).

4. Fig. 1e: CTB-488-labeled neurons look smaller than CTB-647-labeled neurons.

We thank the reviewer for pointing out this caveat. We performed additional experiments to quantify and compare the soma sizes of DRG neurons labeled by CTB-488 and CTB-647 side by side. We found that 488 and 647 CTB beads indeed have slightly different preferences in soma diameters (Supplemental Figure 1). However, since we haven't made any size comparison between the two colors of CTBs throughout the manuscript, we do not think this caveat will affect our finding or conclusions here. Nonetheless, we have added this technical detail in the method section (page 36, line 300-302)

Supplemental Figure 1. CTB labeled DRG soma size distribution from mice with CTB-647 or CTB-488 injected into iWAT (301 neurons from 3 mice for CTB-488, and 379 neurons from 3 mice for CTB-647).

5. Please analyze diameters/sizes of AAV9 infected neurons. Is it possible that AAV9 preferentially labels certain population of sensory neurons?

We have now added cell size quantification for both AAV9 and ROOT labeled neurons (Extended Data Fig. 4j, k) in the revised manuscript. We do not observe a significant difference in the soma sizes between AAV9 and ROOT labeled neurons.

6. I wonder if you saw sex differences in somatosensory innervations of adipose tissue (iWAT). Male and female mice have different body weights and fat ratios.

Thank you for reminding us of this important variable. We have now included the characterization of sensory innervation of adipose tissues in female mice using both anterograde labeling (Extended Data Fig. 1c) and retrograde labeling (Extended Data Fig. 2j, k). Specifically, besides inguinal WAT, we also characterize periovarian white adipose tissue, a female equivalent of the epididymal fat WAT in male. We observed similarly robust sensory innervation of adipose tissues in female mice.

Minor

Fig. 4i: The sample size of Cre- group is small (n=5), compared to other groups (n=8-10).

We have now added more animals to the food intake measurement (currently Extended Data Fig. 8e). We now have N=11 for the Cre- group and N=13 for the Cre+ group.

Line 64: It is stated “pan-DRG Cre transgenic mice (such as Pirt-Cre, Scn10a-Cre) also significantly target sympathetic neurons”. What about Advillin-Cre?

The expression profile of Advillin has been thoroughly investigated in a previous study⁶. The authors found that Advillin is expressed throughout the mouse autonomic neuronal lineage using Advillin antibody, Advillin-EGFP, and Advillin-CreERT2 mice lines. We have now added Advillin-CreERT2 in the main text “...pan-DRG Cre transgenic mice (such as Pirt-Cre, Scn10a-Cre, Advillin-CreERT2)...” (page 4, line 65).

Referee #2 (Remarks to the Author):

The sympathetic nervous system is well known to control adipose tissue thermogenesis and lipolysis. The new work by Wang et al. reports that somatosensory innervation to the inguinal adipose tissue regulates its function. The authors first developed new viral and imaging tools to visualize sensory nerves and found that DRG axons from soma projected to the inguinal adipose tissues in mice. Subsequently, the authors demonstrated that selective ablation of sensory nerves led to transcriptional changes in the inguinal adipose tissue, including elevated expression of lipogenic genes and thermogenic genes.

The findings are novel and significant for the following reasons: first, the authors employed new tools to precisely dissect the projection of somatosensory neurons into the inguinal adipose tissues. TH+ nerves in adipose tissues have been viewed as sympathetic nerves, but this work showed that a part of them are somatosensory neurons. Second, the authors provided evidence that somatosensory neurons control the function of adipose tissues in vivo. The latter, however, needs additional evidence - this reviewer's comment focuses on the area.

We appreciate that the reviewer found our study novel and significant.

Comment 1. The authors showed that 6-OHDA treatment blunted the effect of sensory ablation on thermogenic and lipogenic genes. Intriguingly, however, sensory ablation stimulated lipogenic genes rather than the lipolytic pathway, suggesting more complicated regulation than enhanced sympathetic nerves. If sensory ablation caused de-repression of the sympathetic activity, it would have activated lipolysis, rather than lipogenesis. Thus, the authors need to further dissect the underlying mechanism. For instance, the authors may test if norepinephrine levels in WAT or circulation are altered by sensory ablation. Analyzing changes in the cellular composition of WAT (e.g., immune cell infiltration) or activities of tissue-resident macrophages, ILC2, etc. could be another possibility.

We also find the lipogenic gene expression highly intriguing as it had not been widely regarded as the most prominent consequence of sympathetic activity (especially compared to the thermogenic pathway). However, upon literature search, we found that lipogenic gene programs, particularly Fasn and ChREBP β , were also direct targets of sympathetic outflow as shown by previous studies (E. Motillo, ... and J. G. Granneman, *J Lipid Res.*, 2014; A. Guilherme, ... and M. P. Czech, *Cell Rep.*, 2020)^{7,8}. We also found that ablated fat pad has higher pHSL (Extended Data Fig. 7f), suggesting lipolysis was also increased upon sensory ablation. Thus, our data suggests that sensory ablation upregulates both lipogenesis and lipolysis which are downstream of the sympathetic signaling, although likely shifting the balance to more fat accumulation.

In bilateral ablation mice, we did not observe significant changes in heart rate or blood pressure, suggesting systemic sympathetic tone was not altered (Fig. 4h, Extended Data Fig. 8f-h). We also measured fat norepinephrine content by LCMS and found no detectable difference (Extended Data Fig. 8i). These results suggest that either sensory sympathetic interaction happens downstream of adrenergic receptor(s),

Table 1 Expression profiles of immune cells related genes in adipose tissues after unilateral sensory ablation

	Genes	Fold Change	P value
Macrophages	Itgam (Cd11b)	1.08	0.58
	Cd68	1.16	0.32
Inflammatory markers	Tnf	0.95	0.65
	Il1b	0.91	0.74
	Il6	0.88	0.84

or it induces heterogeneous temporal or local responses within the tissue (which may not be detectable by bulk tissue extract and HPLC-MS).

Examining immune cells and inflammatory activities is a good suggestion. We now analyzed our RNAseq data for general adipose inflammation markers (TNFa, IL6, IL1b) and macrophage markers (CD68, CD11c), but did not find significant changes between ablated and control fat pad (Table 1). We could not rule out the possibility of within-tissue heterogeneity or involvement of more heterogeneous macrophage subtypes. However, we think these are not the main focus of this paper and do not have the expertise to examine these possibilities in detail, given the complexity and controversy around this topic (Y. Qiu, ..., and A. Chawla, *Cell*, 2014; K. Fischer, ..., and C. Buettner, *Nat Med.*, 2017; F. Henriques, ..., and M. P. Czech, *Cell Rep.* 2020)⁹⁻¹¹. We, therefore, suggest that we only include the above Table in the response to the reviewers.

All above points are incorporated in the revised manuscript (result: page 8 line 159-177; discussion page 9-10, line 196-210).

Comment 2. Beige cells are often clustered near the nerve terminals, and the distribution varies on the location of adipose tissues. The authors should provide a comprehensive view of inguinal adipose tissues with a logistic relationship between emerged beige adipocytes and sensory projection.

We have now included immunolabeling against UCP1 in adipose tissues after sensory ablation (Extended Data Fig. 7g) in the revised manuscript. Consistent with the literature, we observed emerged beige adipocytes (UCP1+) in clusters near the lymph node and along the inguinal portion of the fat. However, the mCherry tag we used in the ablation construct is primarily enriched in the soma (Fig. 2d, for the purpose of controlling injection specificity) but poorly transported/expressed in the long-range axon terminals. Therefore, we could not reliably co-localize the emerged UCP1 together with sensory axons in the ablation experiments.

Comment 3. A key question would be to determine the metabolic significance of this neuronal control at a systemic level. The authors should determine the degree to which sensory ablation in adipose tissue affects whole-body energy expenditure and food intake.

We agree it's important to determine the systemic metabolic changes upon fat sensory ablation. We have now added a whole set of characterizations, including food intake, blood pressure, heart rate, temperature preference, and norepinephrine in bulk adipose tissue, to the revised manuscript (Fig. 4g, h, Extended Data Fig. d-i), from which we do not observe significant difference between groups, suggesting sensory ablation in adipose tissue did not lead to general deficits in homeostatic control or the systemic sympathetic tone.

We also now included experiments characterizing how sensory ablated mice responded to a high-fat diet (Extended Data Fig 4j-n, more in detail in the response below to Comment 4), from which we observe minor (not significant) differences in body weight, fasting glucose and insulin levels between groups but profoundly improved glucose tolerance in the ablated mice.

However, we do not have access to a **temperature-controlled** CLAMS system to directly measure the energy expenditure under these conditions (since our body temperature and HFD phenotypes were observed at 30°C thermoneutrality). Moreover, CLAMS experiments are inherently noisy and typically require a large number of matched control/experiment mice (typically >8 per group) to have sufficient

power to detect differences in energy expenditure. Due to technical difficulty, we can perform no more than 2 double-side ablation surgeries per day, limiting the age-matched mice to 5-6/group in most of our experiments. This is another reason we could not directly compare the energy expenditure.

Comment 4. An exciting opportunity for the authors is to see how sensory ablation influences body-weight gain in response to an obesogenic diet, such as a high-fat diet. Given the elevated lipogenic genes and thermogenic genes by sensory ablation, it would be intriguing to see how this leads to changes in body weight and fat mass.

Thank you for this suggestion. We have now included new high-fat diet experiments (Extended Data Fig. 8i-m). We observed minimal or minor differences in body weight, fasting glucose & insulin levels between groups, but we found that the glucose tolerance was profoundly improved in the ablated mice. This disproportional change in glucose tolerance and body weight remarkably resembles the phenotype reported in PRDM16 transgenic mice (a well-established beige fat model)^{12,13}, suggesting the emerging beige fat could be the underlying mechanism of our GTT phenotype.

Referee #3 (Remarks to the Author):

In a manuscript entitled: "The role of somatosensory innervation of adipose tissue" Wang et al. propose that beige fat-innervating sensory neurons modulate adipocyte function by acting as a brake on the sympathetic system.

This is an interesting ms exploring somatosensory innervation of adipose tissue. By acting as a brake on the sympathetic system, the authors show that this type of innervation is an important regulator of adipose tissue function and cellular composition. Over all, the experiments appear well executed with relevant controls. However, there are still some issues that needs to be addressed.

We greatly appreciate the encouragement from the reviewer.

Comments

With the work of the late Timothy J Bartness in mind, it is perhaps a bit oversimplified to state in the Abstract that: "The canonical view is that circulating hormones secreted by the fat convey metabolic state to the brain, which integrates peripheral information and regulates adipocyte function through noradrenergic sympathetic output." Especially relevant in this context is his work using the viral tract tracer, the HI29 strain of the herpes simplex virus-1, he discovered for the first time the sensory pathway by which WAT connects with and informs the central nervous system of events occurring in the fat pad Ref 16 (<https://doi.org/10.1152/ajpregu.90786.2008>). Thus, a link between adipose tissue and the CNS, both efferent and afferent, not dependent on circulating hormones has been reported previously. It would be appropriate to acknowledge this fact a bit more clearly.

We thank the reviewer for prompting this important acknowledgment. We absolutely appreciate the pioneering work from Dr. Bartness on adipose innervation. We changed the statement in the abstract to "The mainstream view..." (Page 2, line 11). We have also included this reference in the introduction, expanded the background, adding "...Although pioneering work using herpes viral tracing has elegantly mapped the central projection of fat afferents..." (page 3, line 38-39) and modified the discussion (page 9, line 197-198) to highlight his earlier work.

The in vivo experiments have been performed using C57BL/J mice, an inbred mice strain prone to develop insulin resistance, obesity, and type 2 diabetes. The authors should make it clear that this is a limitation to their study. We don't know if the findings reported can be generalized or if they are a consequence of this propensity. One way around this problem would be to also perform some of the key experiments using a mice strain less prone to develop these conditions such as A/J.

We thank the reviewer for pointing out this important limitation. We have now included it as a limitation of our study as "it remains to be tested if sensory innervation undergoes a similar process, especially in different strains of mice as well as across species." (page 10, line 209-210).

The authors should in greater detail discuss how central sympathetic out-flow is regulated in response to somatosensory input from WAT. The rabies or herpes virus mediated methodology of retrograde tracing is trans-synaptic and will aid in identifying regions in the CNS that are responsible for

sympathetic out-flow regulation. Have the authors considered to use this methodology to also map CNS regulation of sympathetic out-flow?

As the reviewer pointed out, Dr. Bartness's earlier work using HSV and PRV for tracing fat innervation back to the brain/CNS was highly detailed and comprehensive. Instead of repeating this elegant work, we are currently focusing on characterizing these neural substrates with functional assays based on the anatomical blueprint^{14,15}. While we agree that understanding the upstream targets would be highly informative, we think direct experimental attempts to map the brain circuit are beyond the scope of this paper. We now included these references and future plans in the discussion.

How specific is the regulation evoked by somatosensory input from WAT? Are there any systemic consequences of ablating sensory afference from WAT? Changes in systemic blood pressure? Heart rate? Systemic glucose tolerance?

We thank the reviewer for raising these important questions regarding systemic physiological changes. We have now included measurement of blood pressure (Extended Data Fig. 8f-h), heart rate (Fig. 4h), and temperature sensitivity (Fig. 4g). We found no significant difference between the control and sensory ablation groups, suggesting sensory ablation in adipose tissue did not lead to general deficits in homeostatic control or the systemic sympathetic tone.

Moreover, in response to another reviewer's suggestion, we have now included a high-fat diet challenge for mice that received sensory ablation (Extended Data Fig. 8i-m). We observe minimal or minor differences in body weight or insulin level after the high-fat diet. But we found that the glucose tolerance was markedly improved in the ablated mice. This disproportional change in glucose tolerance and body weight remarkably resembles the phenotype reported in PRDM16 transgenic mice (a well-established beige fat model)^{12,13}, suggesting the emerging beige fat could be the underlying mechanism of our GTT phenotyping.

The authors need to investigate possible changes in global gene expression not only in ipsi- and contralateral fat pads but also in intraabdominal WAT and interscapular BAT to depict any changes at these sites that might be due to altered sympathetic out-flow in response to ablation of somatosensory afference.

We have quantified the gene expression levels by quantitative PCR in both epididymal WAT (eWAT) and interscapular BAT (iBAT) for both unilateral and bilateral sensory ablation experiments (Extended Data Fig. 6a-b, e-f, Extended Data Fig. 8a-b), from which we did not observe significant changes, indicating our fat-specific fat sensory ablation is only affect local sympathetic effect without altering systemic sympathetic tones.

References

1. Jänig, W. Central organization of somatosympathetic reflexes in vasoconstrictor neurones. *Brain Res* 87, 305–312 (1975).
2. LONGHURST, J. C., TJEN-A-LOOI, S. C. & FU, L. Cardiac Sympathetic Afferent Activation Provoked by Myocardial Ischemia and Reperfusion. *Ann Ny Acad Sci* 940, 74–95 (2001).
3. Liu, S. *et al.* Somatotopic Organization and Intensity Dependence in Driving Distinct NPY-Expressing Sympathetic Pathways by Electroacupuncture. *Neuron* 108, 436–450.e7 (2020).
4. Liu, S. *et al.* A neuroanatomical basis for electroacupuncture to drive the vagal–adrenal axis. *Nature* 598, 641–645 (2021).
5. Morelli, C. *et al.* Identification of a population of peripheral sensory neurons that regulates blood pressure. *Cell Reports* 35, 109191 (2021).
6. Hunter, D. V. *et al.* Advillin Is Expressed in All Adult Neural Crest-Derived Neurons. *Eneuro* 5, EnorepinephrineURO.0077-18.2018 (2018).
7. Mottillo, E. P. *et al.* Coupling of lipolysis and de novo lipogenesis in brown, beige, and white adipose tissues during chronic β 3-adrenergic receptor activation. *J Lipid Res* 55, 2276–2286 (2014).
8. Guilherme, A. *et al.* Control of Adipocyte Thermogenesis and Lipogenesis through γ 3-Adrenergic and Thyroid Hormone Signal Integration. *Cell Reports* 31, 107598–107598 (2020).
9. Qiu, Y. *et al.* Eosinophils and Type 2 Cytokine Signaling in Macrophages Orchestrate Development of Functional Beige Fat. *Cell* 157, 1292–1308 (2014).
10. Fischer, K. *et al.* Alternatively activated macrophages do not synthesize catecholamines or contribute to adipose tissue adaptive thermogenesis. *Nat Med* 23, 623–630 (2017).
11. Henriques, F. *et al.* Single-Cell RNA Profiling Reveals Adipocyte to Macrophage Signaling Sufficient to Enhance Thermogenesis. *Cell Reports* 32, 107998–107998 (2020).
12. Seale, P. *et al.* Prdm16 determines the thermogenic program of subcutaneous white adipose tissue in mice. *J Clin Invest* 121, 96–105 (2011).
13. Ikeda, K. *et al.* UCP1-independent signaling involving SERCA2b-mediated calcium cycling regulates beige fat thermogenesis and systemic glucose homeostasis. *Nat Med* 23, 1454–1465 (2017).
14. Song, C. K., Schwartz, G. J. & Bartness, T. J. Anterograde transneuronal viral tract tracing reveals central sensory circuits from white adipose tissue. *Am J Physiology-regulatory Integr Comp Physiology* 296, R501–R511 (2009).
15. Ryu, V. & Bartness, T. J. Short and long sympathetic-sensory feedback loops in white fat. *Am J Physiology-regulatory Integr Comp Physiology* 306, R886–R900 (2014).

Reviewer Reports on the First Revision:

Referees' comments:

Referee #1 (Remarks to the Author):

The authors have addressed my concerns with substantial experimental efforts. The scientific rigor has been greatly improved by adding additional characterization of the labeled neurons. The authors also tested systemic sympathetic functions and examined the effects of metabolic challenges.

Referee #2 (Remarks to the Author):

The authors provided new data and explanations that addressed this reviewer's comments. It is intriguing that sensory ablation profoundly improved glucose tolerance.

Re: Comment 2. The authors mentioned that new images of adipose tissue immunolabeling against UCP1 were shown in Extended Data Fig. 7g But this is not found. The authors should revise the figures and manuscript accordingly.

Referee #3 (Remarks to the Author):

My concerns have been adequately addressed.